# Functional Regularization for Reinforcement Learning via Learned Fourier Features

**Alexander C. Li**
Carnegie Mellon University
alexanderli@cmu.edu

**Deepak Pathak**
Carnegie Mellon University
dpathak@cs.cmu.edu

## Abstract

We propose a simple architecture for deep reinforcement learning by embedding inputs into a learned Fourier basis and show that it improves the sample efficiency of both state-based and image-based RL. We perform infinite-width analysis of our architecture using the Neural Tangent Kernel and theoretically show that tuning the initial variance of the Fourier basis is equivalent to *functional* regularization of the learned deep network. That is, these learned Fourier features allow for adjusting the degree to which networks underfit or overfit different frequencies in the training data, and hence provide a controlled mechanism to improve the stability and performance of RL optimization. Empirically, this allows us to prioritize learning low-frequency functions and speed up learning by reducing networks' susceptibility to noise in the optimization process, such as during Bellman updates. Experiments on standard state-based and image-based RL benchmarks show clear benefits of our architecture over the baselines[1].

## 1 Introduction

Most popular deep reinforcement learning (RL) approaches estimate either a value or Q-value function under the agent's learned policy. These functions map points in the state or state-action space to expected returns, and provide crucial information that is used for improving the policy. However, optimizing these functions can be difficult, since there are no ground-truth labels to predict. Instead, they are trained through bootstrapping: the networks are updated towards target values calculated with the same networks being optimized. These updates introduce noise that accumulates over repeated iterations of bootstrapping, which can result in highly inaccurate value or Q-value estimates [44, 46]. As a result, these RL algorithms may suffer from lower asymptotic performance or sample efficiency.

Most prior work has focused on making the estimation of target values more accurate. Some examples include double Q-learning for unbiased target values [16, 47], or reducing the reliance on the bootstrapped Q-values by calculating multi-step returns with TD($\lambda$) [41]. However, it is impossible to hope that the noise in the target values estimated via bootstrapping will go to zero, because we cannot estimate the true expectation over infinite rollouts in practical setups. Hence, we argue that it is equally important to also regularize the function (in this case, the deep Q-network) that is fitting these noisy target values.

Conventional regularization methods in supervised learning can be associated with drawbacks in RL. Stochastic methods like dropout [40] introduce more noise into the process, which is tolerable when ground-truth labels are present (in supervised learning) but counterproductive when bootstrapping. An alternative approach is early stopping [50], which hurts sample efficiency in reinforcement learning because a minimum number of gradient steps is required to propagate value information backwards to early states. Finally, penalty-based methods like $L_1$ [45] and $L_2$ regularization [17, 22] can help in

---

[1]Code available at https://github.com/alexlioralexli/learned-fourier-features

35th Conference on Neural Information Processing Systems (NeurIPS 2021).

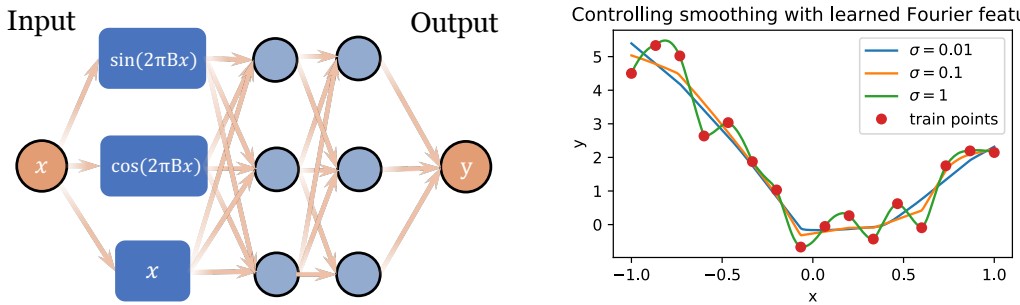

Figure 1: **Left:** the proposed learned Fourier feature (LFF) architecture. $B$ is a matrix trained through backpropagation, and is used to create a learned set of Fourier features. The network then passes the Fourier features through alternating linear layers and ReLU nonlinearities, as is done in vanilla MLPs. **Right:** tuning the initialization variance of $B_{ij} \sim \mathcal{N}(0, \sigma^2)$ controls the rate at which target frequencies are learned. Higher $\sigma$ fits higher frequencies faster, while lower $\sigma$ smooths out noise. This architecture can be used as functional regularization for a Q-function $Q : \mathcal{S} \times \mathcal{A} \to \mathbb{R}$, value function $V : \mathcal{S} \to \mathbb{R}$, policy $\pi : \mathcal{S} \to \mathbb{R}^{\dim(\mathcal{A})}$, or model $T : \mathcal{S} \times \mathcal{A} \to \mathcal{S}$.

RL [26], but regularizing the network in weight space does not disentangle noise from reward signal and could make it difficult to learn the true Q-function. This leads us to ask: what is the right way to regularize the RL bootstrapping process?

We suggest that the impact of target value noise can be better reduced by *frequency-based functional regularization*: direct control over the frequencies that the network tends to learn first. If the target noise consists of higher frequencies than the true target values, discouraging high-frequency learning can help networks efficiently learn the underlying Q-function while fitting minimal amounts of noise. In this work, we propose an architecture that achieves this by encoding the inputs with learned Fourier features, which we abbreviate as LFF. In contrast to using fixed Fourier features [34, 42], we train the Fourier features, which helps them find an appropriate basis even in high dimensional domains. We analyze our architecture using the Neural Tangent Kernel [18] and theoretically show that tuning the initial variance of the Fourier basis controls the rate at which networks fit different frequencies in the training data. Thus, LFF's initial variance provides a controlled mechanism to improve the stability and performance of RL optimization (see Figure 1). Tuned to prioritize learning low frequencies, LFF filters out bootstrapping noise while learning the underlying Q-function.

We evaluate LFF, which only requires changing a few lines of code, on state-space and image-space DeepMind Control Suite environments [43]. We find that LFF produces moderate gains in sample efficiency on state-based RL and dramatic gains on image-based RL. In addition, we empirically demonstrate that LFF makes the value function bootstrapping stable even in absence of target networks, and confirm that most of LFF's benefit comes through regularizing the Q-network. Finally, we provide a thorough ablation of our architectural design choices.

## 2 Preliminaries

The reinforcement learning objective is to solve a Markov Decision Process (MDP), which is defined as a tuple $(\mathcal{S}, \mathcal{A}, P, R, \gamma)$. $\mathcal{S}$ and $\mathcal{A}$ denote the state and action spaces. $P(s'|s, a)$ is the transition function, $R(s, a)$ is the reward function, and $\gamma \in [0, 1]$ is the discount factor. We aim to find an optimal policy $\pi^*(a|s)$ that maximizes the expected sum of discounted rewards. Q-learning finds the optimal policy by first learning a function $Q^*(s, a)$ such that $Q^*(s, a) = \mathbb{E}_{s' \sim P(\cdot|s,a)}[R(s, a) + \gamma \max_{a'} Q^*(s', a')]$. The optimal policy is then $\pi^*(a|s) = \arg\max_a Q^*(s, a)$. To find $Q^*$, we repeatedly perform Bellman updates to $Q$, which uses the Q-network itself to bootstrap target values on observed transitions $(s, a, r, s')$ [6]. The most basic way to estimate the target values is $target = r + \gamma \max_{a'} Q(s', a')$, but a popular line of work in RL aims to find more accurate target value estimation methods [12, 16, 32, 41, 47]. Once we have these target value estimates, we update our Q-network's parameters $\theta$ via gradient descent on temporal difference error: $\Delta\theta = -\eta \nabla_\theta (Q_\theta(s, a) - target)^2$. Our focus is on using the LFF architecture to prevent Q-networks from fitting irreducible, high-frequency noise in the target values during these updates.

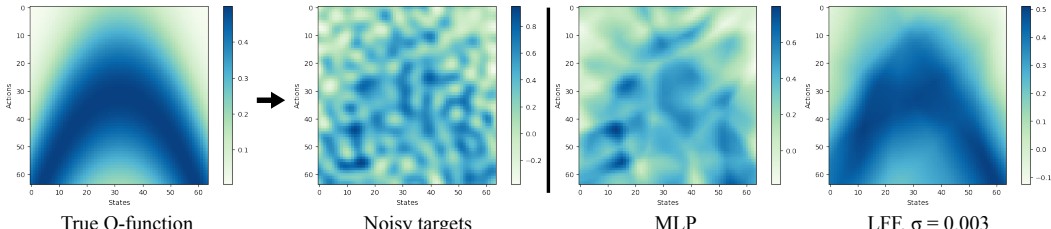

| True Q-function | Noisy targets | MLP | LFF, σ = 0.003 |

Figure 2: **Filtering noise with LFF input embedding.** The bootstrapped targets in RL are a mix between signal and noise. Right side: we fit different networks to these noisy targets and display their predictions. While the MLP overfits, LFF learns the Q-function and ignores almost all of the noise.

## 3 Reinforcement Learning with Learned Fourier Features

We present a visualization of the noisy target value problem in Figure 2. The target value estimates can be noisy due to stochastic transitions, replay buffer sampling, or unintended generalization across state-action pairs due to function approximation [48]. We simulate this in Figure 2 by adding noise to the optimal Q-function of a small gridworld. MLPs are susceptible to fitting the noise, resulting in inaccurate Q-values that could diverge after repeated bootstrapping. In contrast, our LFF architecture controls how quickly low- and high-frequency signals are learned. Tuned properly, LFF filters out the noise and learns the ground truth Q-function almost perfectly (Figure 2, right).

The problem is that MLPs provide no control over how quickly they learn signals of different frequencies. Prior work in computer vision found this a problem when MLPs blurred desired high frequencies in low-dimensional (3-5 dimensions) graphics regression problems [28, 42]. They fixed this blurring problem by transforming the input using a random Fourier feature embedding of the input [34, 42]. Specifically, the idea is to map a low-dimensional input $x$ to an embedding $\gamma(x) = \sin(2\pi Bx) || \cos(2\pi Bx)$, where $B$ is a $d_{\text{fourier}}/2 \times d_{\text{input}}$ matrix, $||$ denotes concatenation, and sin and cos act elementwise. The embedding $\gamma(x)$ directly provides a mix of low- and high-frequency functions a MLP $f_\theta$ can use to learn a desired function. Tancik et al. [42] use this to improve fidelity in coordinate-based graphics problems.

Intuitively, the row vectors $b_i$ are responsible for capturing desired frequencies of the data. If they capture only low frequencies, then the MLP will be biased towards only learning the low-frequency signals in the data. Conversely, if the Fourier features capture sufficient high-frequency features, then a MLP can fit high frequency functions by computing simple nonlinear combinations of the features. In these low-dimensional graphics problems, initializing fixed entries $B_{ij} \sim \mathcal{N}(0, \sigma^2)$ with large $\sigma$ was enough to learn desired high frequency functions; training $B$ did not improve performance.

In the following section, we propose our learned Fourier feature architecture for deep RL, which allows practitioners to *tune the range of frequencies that the network should be biased towards learning*. We propose several key enhancements that help Fourier features learn in high-dimensional environments. Our work uses learned Fourier features to improve deep RL, in contrast to prior work focused on simple environments with fixed, hand-designed Fourier features and linear function approximation [20, 21]. Although we focus on prioritizing low-frequency signals to reduce bootstrap noise, we also present cases in Appendix F.1 where biasing networks towards high-frequency learning with learned Fourier features enables fast convergence and high asymptotic performance in RL.

### 3.1 Learned Fourier Feature Architecture

Standard MLPs can be written as the repeated, alternating composition of affine transformations $L_i(x) = W_i x + b_i$ and nonlinearity $\tau$, which is usually the ReLU $\tau(x) = \max(0, x)$:

$$f_\theta(x) = L_n \circ \tau \circ L_{n-1} \circ \tau \circ \cdots \circ L_1(x) \qquad (1)$$

We propose a novel architecture based on Fourier features, shown in Figure 1. We define a new layer:

$$F_B(x) = \sin(2\pi Bx) || \cos(2\pi Bx) || x \qquad (2)$$

where $B$ is a $d_{\text{fourier}}/2 \times d_{\text{input}}$ matrix and $||$ denotes concatenation. $d_{\text{fourier}}$ is a hyperparameter that controls the number of Fourier features we can learn; increasing $d_{\text{fourier}}$ increases the degree to which the model relies on the Fourier features. Following Tancik et al. [42], we initialize the entries $B_{ij} \sim N(0, \sigma^2)$, where $\sigma^2$ is a hyperparameter. Contrary to prior work, $B$ is a trainable parameter.

The resulting LFF MLP can be written:

$$f_\theta = L_n \circ \tau \circ \cdots \circ L_1 \circ F_B(x) \qquad (3)$$

We can optimize this the same way we optimize a standard MLP, e.g. regression would be:

$$\arg\min_{\theta, B} \sum_{i=1}^{N} (L_n \circ \cdots \circ F_B(x_i) - y_i)^2 \quad (4)$$

We propose two key improvements to random Fourier feature input embeddings [34, 42]: training $B$ and concatenating the input $x$ to the Fourier features. We hypothesize that these changes help preserve information in high-dimensional RL problems, where it is increasingly unlikely that randomly initialized $B$ produces Fourier features well-suited for the task.

**Algorithm 1** LFF PyTorch-like pseudocode.

```
class LFF():
    def __init__(self, input_size, output_size,
                 n_hidden=1, hidden_dim=256,
                 sigma=1.0, f_dim=256):
        # create B
        b_shape = (input_size, f_dim // 2)
        self.B = Parameter(normal(zeros(*b_shape),
                           sigma * ones(*b_shape)))
        # create rest of network
        self.mlp = MLP(in_dims=f_dim + input_size,
                       out_dims=output_size,
                       n_hidden=n_hidden,
                       hidden_dim=hidden_dim)
    def forward(self, x):
        proj = (2 * np.pi) * matmul(x, self.B)
        ff = cat([sin(proj), cos(proj), x], dim=-1)
        return self.mlp.forward(ff)
```

normal: sample from Gaussian with specified mean and std dev; matmul: matrix multiplication; cat: concatenation.

Training $B$ alleviates this problem by allowing the network to discover them on its own. Appendix G shows that training $B$ does change its values, but its variance remains close to the initial $\sigma^2$. This indicates that $\sigma$ is an important knob that controls the network behavior throughout training. Concatenating $x$ to the Fourier features is another key improvement for high dimensional settings. It preserves all the input information, which has been shown to help in RL [38]. We further analyze these improvements in Section 6.4.

## 4 Theoretical Analysis

By providing periodic features (which are controlled at initialization by the variance $\sigma^2$), LFF biases the network towards fitting a desired range of frequencies. In this section, we hope to understand why the LFF architecture provably controls the rate at which various frequencies are learned. While Tancik et al. [42] analyze different weightings of a full set of Fourier basis frequencies, we examine the effect of the initialization variance of randomly initialized Gaussian Fourier features.

We draw upon linear neural network approximations in the infinite width limit, which is known as the neural tangent kernel approach [18]. This approximation allows us to understand the training dynamics of the learned neural network output function. While NTK analysis has been found to diverge from real-world behavior in certain cases, particularly for deeper convolutional neural networks [2, 10], it has also been remarkably accurate in predicting directional phenomena [4, 5, 42]. We provide background on the NTK in Section 4.1, and discuss the connection between the eigenvalues of the NTK kernel matrix and the rate at which different frequencies are learned in Section 4.2. We then analyze the NTK and frequency learning rate for Fourier features in networks with 2 layers (Section 4.3) or more (Section 4.4).

### 4.1 Neural Tangent Kernel

We can approximate a neural network using a first-order Taylor expansion around its initialization $\theta_0$:

$$f_\theta(x) \approx f_{\theta_0}(x) + \nabla_\theta f_{\theta_0}(x)^\top (\theta - \theta_0) \qquad (5)$$

The Neural Tangent Kernel [18] line of analysis makes two further assumptions: $f_\theta$ is an infinitely wide neural network, and it is trained via gradient flow. Under the first condition, a trained network stays very close to its initialization, so the Taylor approximation is good (the so-called "lazy training" regime [8]). Furthermore, $f_{\theta_0}(x) = 0$ in the infinite width limit, so our neural network is simply a linear model over features $\phi(x) = \nabla_\theta f_{\theta_0}(x)$. This gives rise to the kernel function:

$$k(x_i, x_j) = \langle \nabla_\theta f_{\theta_0}(x_i), \nabla_\theta f_{\theta_0}(x_j) \rangle \qquad (6)$$

The kernel function $k$ is a similarity function: if $k(x_i, x_j)$ is large, then the predictions $f_\theta(x_i)$ and $f_\theta(x_j)$ will tend to be close. This kernel function is deterministic and does not change over the course of training, due to the infinite width assumption. If we have $n$ training points $(x_i, y_i)$, $k$ defines a PSD kernel matrix $K \in \mathbb{R}_+^{n \times n}$ where each entry $K_{ij} = k(x_i, x_j)$. When we train this infinite width

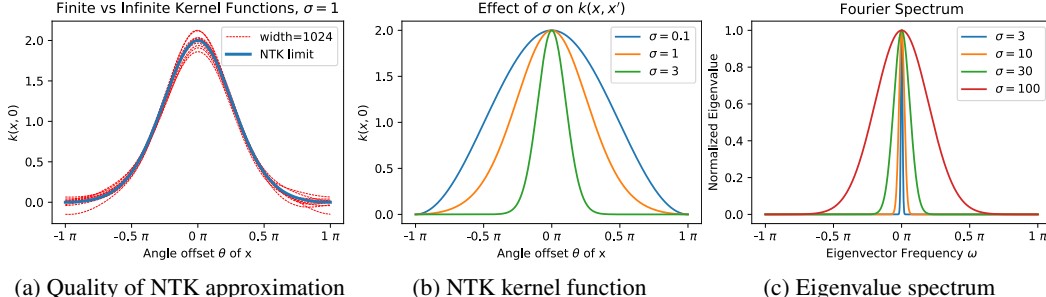

(a) Quality of NTK approximation      (b) NTK kernel function      (c) Eigenvalue spectrum

Figure 3: **NTK of the 2-layer Fourier feature model.** We plot the NTK $k(x, 0)$ for points $x$ that lie on the unit circle. In (a) and (b), $\theta \in [-\pi, \pi]$ denotes the offset from a reference point with $\theta_0 = 0$. Note that our NTK is shift invariant, so these figures are valid for any reference point $\theta_0$. **Left:** we compare the NTK infinite-width limit to the kernel function of 10 randomly initialized 2-layer Fourier feature networks with width 1024. The NTK limit is quite accurate for realistically wide networks. **Middle:** NTK kernel function $k$ for varying settings of the Fourier feature variance $\sigma^2$. Larger $\sigma$ enables sharp, local learning, while smaller $\sigma$ induces smoother function learning. **Right:** The y-axis shows eigenvalues of the NTK kernel matrix $K$, and the x-axis indicates the frequency of the corresponding eigenvector. High frequencies are not fit with low $\sigma$, since their eigenvalues vanish. Increasing $\sigma$ increases the higher frequencies' eigenvalues, so they can be learned faster.

neural network with gradient flow on the squared error, we precisely know the model output at any point in training. At time $t$, we have training residual:

$$f_{\theta_t}(x) - y = e^{-\eta K t}(f_{\theta_0}(x) - y) \tag{7}$$

where $f_{\theta_t}(x)$ is the column vector of model predictions for all $x_i$, and $y$ is the column vector of stacked training labels (see Appendix A.1 for proof sketch). Eq. 7 is critical because it describes how different components of the training loss decrease over time. Section 4.2 will build on this result and examine the training residual in the eigenbasis of the kernel matrix $K$. This analysis will reveal that each frequency present in the labels $y$ will be learned at its own rate, determined by $K$'s eigenvalues.

### 4.2 Eigenvalues of the LFF NTK

Consider applying the eigendecomposition of $K = Q\Lambda Q^*$ to Eq. 7, noting that $e^{-\eta K t} = Q e^{-\eta \Lambda} Q^*$ as the matrix exponential is defined $e^X := \sum_{k=0}^{\infty} \frac{1}{k!} X^k$, so $Q$ and $Q^*$ will repeatedly cancel in the middle of $X^k$ since $Q$ is unitary.

$$Q^*(f_{\theta_t}(x) - y) = e^{-\eta \Lambda t} Q^*(f_{\theta_0}(x) - y) \tag{8}$$

Note that the $i$th component of the residual $Q^*(f_{\theta_t}(x) - y)$ decreases with rate $e^{-\eta \lambda_i}$.

Consider the scenario where the training inputs $x_i$ are evenly spaced on the $d$-dimensional sphere $\mathbb{S}^{d-1}$. When $k$ is isotropic, which is true for most networks whose weights are sampled isotropically, the kernel matrix $K$ is circulant (each row is a shifted version of the row above). In this special case, $K$'s eigenvectors correspond to frequencies from the discrete Fourier transform (DFT), and the corresponding eigenvalues are the DFT values of the first row of $K$ (see Appendix A.2). Combining this fact with Eq. 8, where we looked at the residual in $K$'s eigenbasis, shows that each frequency in the targets is learned at its own rate, determined by the eigenvalues of $K$. For a ReLU MLP, these eigenvalues decay approximately quadratically with the frequency [4]. This decay rate is slow, so MLPs often fit undesirable medium and high frequency signals. **We hypothesize that LFF controls the frequency-dependent learning rate by tuning the kernel matrix $K$'s eigenvalues.** We examine LFF's kernel matrix eigenvalues in Sections 4.3 and 4.4 and verify this hypothesis.

### 4.3 NTK Analysis of 2-layer Network with Fourier Features

To simplify our LFF NTK analysis, we consider a two layer neural network $f : \mathbb{R}^d \to \mathbb{R}$:

$$f(x) = \sqrt{\frac{2}{m}} W^\top \begin{bmatrix} \sin(Bx) \\ \cos(Bx) \end{bmatrix} \tag{9}$$

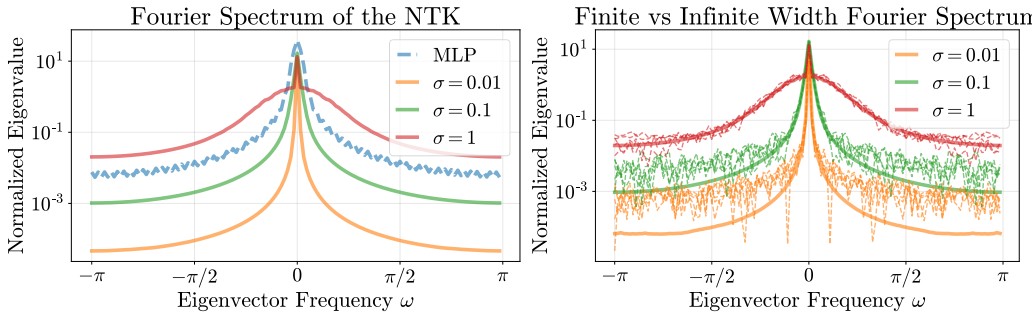

Figure 4: Left: we compare the NTK eigenvalue spectrum (which determines the frequency-specific learning rate) of deep networks with Fourier features to those of a vanilla MLP. Right: we initialize finite-width LFF networks with 2 hidden layers of 1024 units each and compare their kernels (dashed) to the corresponding NTK infinite-width limit (solid). We find that the NTK is accurate (note the log scale) and that decreasing $\sigma$ indeed results in lower convergence rates for higher frequencies.

where each row of $B$ is a vector $b_i^\top \in \mathbb{R}^{1 \times d}$, and there are $m$ rows of $B$. $W_i \sim \mathcal{N}(0, 1)$ and $B_{ij} \sim \mathcal{N}(0, \sigma^2)$, where $\sigma$ is a hyperparameter. Note that concatenating $x$ is omitted for this two-layer model. This is because any contribution from concatenation goes to zero as we increase the layer width $m$. Lemma 1 determines an analytical expression for the LFF kernel function $k(x, x')$.

**Lemma 1.** *For $x, x' \in \mathbb{S}^{d-1}$ with angle $\theta = \cos^{-1}(x^\top x')$, we have the NTK kernel function:*

$$k(x, x') = \left( 2 - \frac{\|x - x'\|_2^2}{2} \right) \exp \left\{ -\frac{\sigma^2}{2} \|x - x'\|_2^2 \right\} \tag{10}$$

Proof: see Appendix A.3. This closed form expression for $k(x, x')$ elucidates several desirable properties of Fourier features. $\sigma$ directly controls the rate of the exponential decay of $k$, which is the similarity function for points $x$ and $x'$. For large $\sigma$, $k(x, x')$ rapidly goes to 0 as $x$ and $x'$ get farther apart, so their labels only affect the learned function output in a small local neighborhood. This intuitively corresponds to high-frequency learning. In contrast, small $\sigma$ ensures $k(x, x')$ is large, even when $x$ and $x'$ are relatively far apart. This induces smoothing behavior, inhibiting high-frequency learning. We plot the NTK for varying levels of $\sigma$ in Figure 3(b) and show that $\sigma$ directly controls the frequency learning speed in Figure 3(c). Figure 3(a) also verifies that the NTK limit closely matches the empirical behavior of realistically sized networks at initialization.

### 4.4 NTK of Deeper Networks

Figure 3(c) shows that larger initialization variance $\sigma^2$ corresponds to larger eigenvalues for high frequencies in the 2-layer model. This matches empirical results that small $\sigma$ leads to underfitting and large $\sigma$ leads to overfitting [42]. However, Figure 3(c) indicates that only extremely large $\sigma$, on the order of $10^2 - 10^3$, result in coverage of the high frequencies. This contradicts Tancik et al. [42], who fit fine-grained image details with $\sigma \in [1, 10]$. We suggest that the 2-layer model, even though it accurately predicts the directional effects of increasing or decreasing $\sigma$, fails to accurately model learning in realistically sized networks. Manually computing the kernel functions of deeper MLPs with Fourier feature input embeddings is difficult. Thus, we turn to Neural Tangents [30], a library that can compute the kernel function of any architecture expressible in its API.

We initialize random Fourier features of size 1024 with different variances $\sigma^2$ and build an infinite-width ReLU MLP on top with 3 hidden layers using the Neural Tangents library. As in Figure 3, we take input data $x$ evenly spaced on the 2D unit circle and evaluate the corresponding kernel function $k(x, 0)$ between the point $(1, 0)$ and the point $x = (\cos\theta, \sin\theta)$. Figure 4 shows the eigenvalues of Fourier features and vanilla MLPs in this scenario. We see the same trend, where increasing $\sigma$ leads to larger eigenvalues for higher frequencies, as we saw in Figure 3. Furthermore, Figure 4 shows that this trend also holds for the exact finite-width architectures that we use in our experiments (Section 6). These results now reflect the empirical behavior where $\sigma \in [1, 10]$ results in high frequency learning. This indicates that deeper networks are crucial for understanding the behavior of Fourier features.

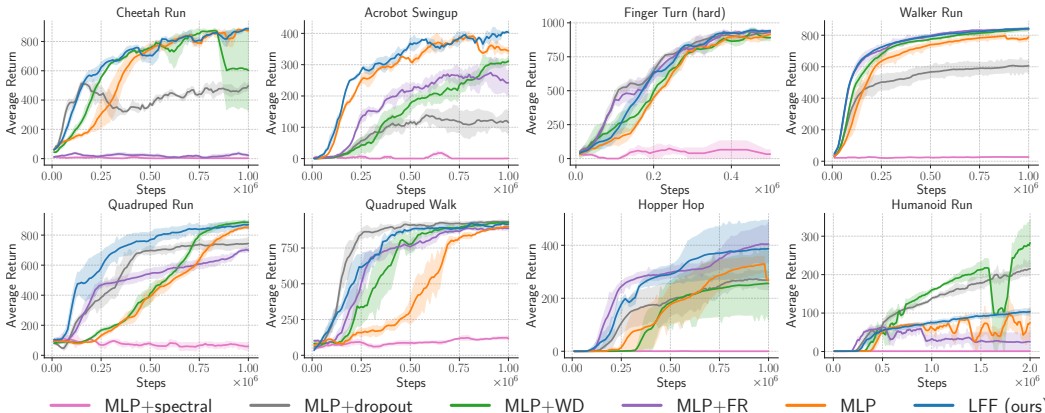

Figure 5: **Off-policy State-based Evaluation**: Soft Actor Critic (SAC) experiments on 8 DM Control environments. We emphasize that these results are produced using the same hyperparameters (e.g. learning rate, Polyak averaging parameter, and batch size) tuned for MLPs. These results show that plugging in our LFF architecture can yield more sample-efficient learning on most environments.

## 5  Experimental Setup

We treat the learned Fourier feature network as a drop-in replacement for MLP and CNN architectures. We show that just adding Fourier features improves the performance of *existing* state-of-the-art methods on *existing* standard benchmark environments from DeepMind Control Suite [43]. We will release the code, **which involves only changing a few lines of code in existing RL algorithms**.

**State-based LFF Architecture Setup**    We use soft actor-critic (SAC), an entropy-regularized off-policy RL algorithm [14], to learn 8 environments from the DeepMind Control Suite [43]. We keep the default hyperparameters fixed, varying only the architecture for the policy and Q-function. Our LFF architecture uses our learnable Fourier feature input layer, followed by 2 hidden layers of 1024 units. We use Fourier dimension $d_{\text{fourier}}$ of size 1024. We initialize the entries of our trainable Fourier basis with $B_{ij} \sim \mathcal{N}(0, \sigma^2)$, with $\sigma = 0.01$ for all environments except Cheetah, Walker, and Hopper, where we use $\sigma = 0.001$. To make the parameter count roughly equal, we compare against an MLP with three hidden layers. The first MLP hidden layer is slightly wider, about 1100 units, to compensate for the extra parameters in LFF's first layer due to input concatenation. Learning curves are averaged over 5 seeds, with the shaded region denoting 1 standard error.

**Image-based LFF Architecture Setup**    We test image-based learning on 4 DeepMind Control Suite environments [43] with SAC + RAD [24], which uses data augmentation to improve the sample efficiency of image-based training. The vanilla RAD architecture, which uses the convolutional architecture from Srinivas et al. [39], is denoted as "CNN" in Figure 6. To apply LFF to images, we observe that computing $Bx$ at each pixel location is equivalent to a 1x1 convolution without bias. This 1x1 convolution maps the the RGB channels at each pixel location from 3 dimensions to $d_{\text{fourier}}/2$ channels. We then compute the $\sin$ and $\cos$ of those channels and concatenate the original RGB values, so our image goes from $H \times W \times 3$ to a $H \times W \times (d_{\text{fourier}} + 3)$ embedding. The 1x1 conv weights are initialized from $\mathcal{N}(0, \sigma^2)$ with $\sigma = 0.1$ for Hopper and Cheetah and $\sigma = 0.01$ for Finger and Quadruped. As we did in the state-based setup, we make the CNN baseline fair by adding an additional 1x1 convolution layer at the beginning. This ensures that the "CNN" and "CNN+LFF" architectures have the same parameter count, and that performance gains are solely due to LFF.

## 6  Results

We provide empirical support for the approach by investigating the following questions:

1. Does LFF improve the sample efficiency of off-policy state-based or image-based RL?
2. Do learned Fourier features make the Bellman update more stable?
3. Does LFF help more when applied to the policy or the Q-function?
4. Ablation: How important is input concatenation or training the Fourier basis $B$?

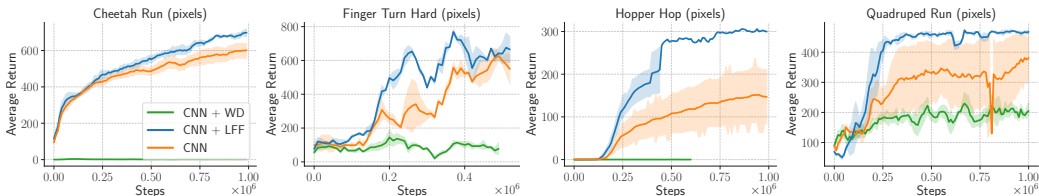

Figure 6: **Off-policy Image-based Evaluation**: SAC experiments on learning 4 DMControl environments from pixels. LFF can yield dramatic improvements in sample-efficiency over CNNs.

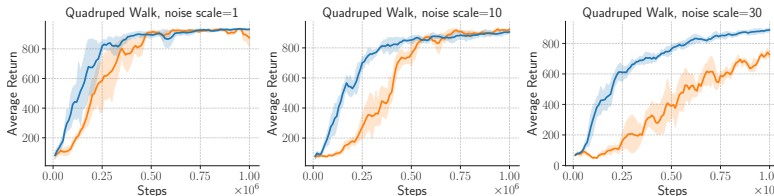

Figure 7: **State-based with Added Noise**: We add zero-mean Gaussian noise to the targets. As the standard deviation of the added noise increases, LFF maintains its performance better than MLPs.

## 6.1 LFF Architecture for Off-policy RL

We show the results of using the LFF architecture for state-based RL with SAC in Figure 5. LFF does clearly better than MLPs in 6 out of 8 environments, and slightly better in the remaining 2. Figure 6 shows even stronger results on image-based RL with SAC and RAD. This is especially promising because these results use the hyperparameters that were tuned for the MLP or CNN baseline. We find that the return consistently starts increasing much earlier with LFF. We hypothesize that LFF reduces noise propagation due to bootstrapping, so less data is required to overcome incorrect targets. SAC can use these more accurate Q-values to quickly begin exploring high-reward regions of the MDP.

For the state-space experiments, we also test several baselines:

- MLP with weight decay, tuned over the values $\{10^{-3}, 3 \times 10^{-4}, 10^{-4}, 3 \times 10^{-4}, 10^{-5}\}$. Weight decay helps learning in most environments, but it can hurt performance (Acrobot, Hopper) or introduce instability (Cheetah). Weight decay strong enough to reduce overfitting may simultaneously bias the Q-values towards 0 and cause underestimation bias.

- MLP with dropout [40]. We add a dropout layer after every nonlinearity in the MLP. We search over $[0.05, 0.2]$ for the drop probability, and find that lower is better. Dropout does help in most environments, although occasionally at the cost of asymptotic performance.

- MLP with functional regularization [31]. Instead of using a target network to compute target values, we use the current Q-network, but regularize its values from diverging too far from the Q-values calculated by a lagging snapshot of the Q-network.

- MLP with spectral normalization [13]. We add spectral normalization to the second-to-last layer of the network, as is done in [13], but find that this works very poorly. It is likely necessary to tune the other hyperparameters (learning rate, target update frequency, Polyak averaging) in order to make spectral normalization work.

Overall, LFF consistently ranks around the top across all of the environments. It can be combined with weight decay, dropout, or functional regularization for more gains, and has a simple plug-and-play advantage because a single set of parameters works over all environments.

## 6.2 Do learned Fourier features improve the stability of the Bellman updates?

Our key hypothesis is that standard ReLU MLP Q-functions tend to fit the noise in the target values, introducing error into the Q-function at some $(s, a)$. Bootstrapping with this incorrect Q-value to calculate target values for other $(s', a')$ yields even noisier targets, propagates the error to other states, and causes instability or divergence (see Appendix D for more details). To further test whether LFF solves this problem by filtering out the noise, we train a SAC agent on state-based DMControl

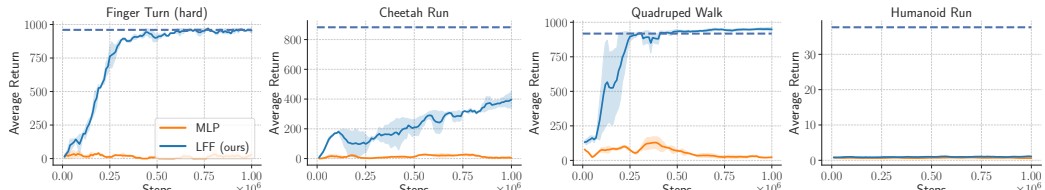

Figure 8: **Effect of LFF on Stability of Bootstrapping**: We train SAC, foregoing a target network, by bootstrapping directly from the Q-network being trained. The dashed line shows the LFF performance with a target network after 1M steps. We find that the LFF network is remarkably stable, and even learns faster on Quadruped Walk than when using target networks. However, LFF fails to learn on Humanoid, indicating that higher dimensional problems still pose problems.

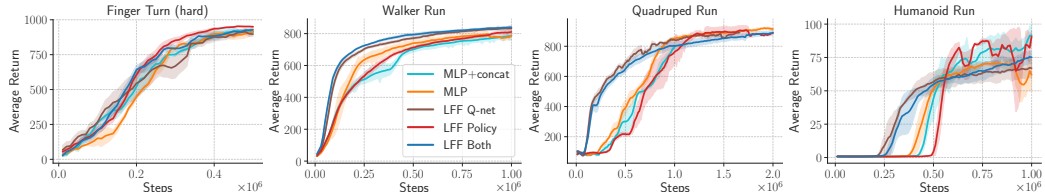

Figure 9: **LFF Policy vs Q-function.** Walker and Quadruped results indicate that only using LFF for the Q-network is just as good as using LFF for both networks. In contrast, using LFF for the policy network is about as bad as the MLP baseline. This suggests that LFF primarily improves off-policy learning by reducing noise in the Q-network optimization.

environments with either of the following modifications: testing the Q-function's robustness by adding Gaussian noise to the targets, or removing target networks altogether.

**Gaussian noise added to targets** In each bootstrap step, we add zero-mean Gaussian noise with standard deviation 1, 10, or 30 to the targets. LFF maintains higher performance even at significant noise levels, indicating that it is more robust to bootstrap noise. Full results are in Figure 12.

**No target network** Target networks, updated infrequently, slow down the propagation of noise due to bootstrapping [29]. LFF fits less noise to begin with, so it should work even when the target network is omitted. Here, we bootstrap directly from the network being trained. Figure 8 shows that MLPs consistently fail to learn on all environments in this setting, while the LFF architecture still performs well, except when the problem is very high dimensional. LFF even manages to learn faster in Quadruped Walk than it does when using a target network, since there is no longer Polyak averaging [25] with a target to slow down information propagation. Omitting the target network allows us to use updated values for $Q_\theta(s', a')$, instead of stale values from the target network. This result is in line with recent work that achieves faster learning by removing the target network and instead penalizing large changes to the Q-values [37].

Overall, Figure 7 and 8 validate our theoretical claims that LFF controls the effect of high-frequency noise on the learned function, and indicates that LFF successfully mitigates bootstrap noise in most cases. Tuning the SAC hyperparameters should increase LFF sample efficiency even further, since we can learn more aggressively when the noise problem is reduced.

### 6.3 Where Do Learned Fourier Features Help?

In this section, we confirm that our LFF architecture improves RL performance by primarily preventing the Q-network from fitting noise. We train state-based SAC with an MLP policy and LFF Q-network (LFF Q-net in Figure 9), or with an LFF policy and MLP Q-network (LFF Policy). Figure 9 shows that solely regularizing the Q-network is sufficient for LFF's improved sample efficiency. This validates our hypothesis that the Bellman updates remain noisy, even with tricks like double Q-networks and Polyak averaging, and that LFF reduces the amount of noise that the Q-network accumulates. These results suggest that separately tuning $\sigma$ for the Q-network and policy networks may yield further improvements, as they have separate objectives. The Q-network should be resilient to noise, while the policy can be fine-grained and change quickly between nearby states. However,

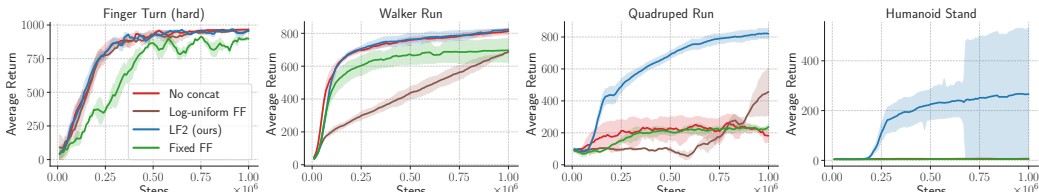

Figure 10: **Ablation analysis**: We train SAC on 4 DMControl environments with three variants of our architecture: LFF, LFF with fixed Fourier features, and LFF without input concatenation. Lower dimensional environments like Finger and Walker are more forgiving for fixed Fourier features or omitting input concatenation. However, Humanoid absolutely requires both modifications to learn.

for simplicity, we use LFF for the Q-networks and the policy networks. Finally, we also train vanilla MLPs where we concatenate the input $x$ to the first layer output. LFF outperforms this variant, confirming that concatenation is not solely responsible for the improved sample efficiency.

## 6.4 Architectural Ablations

LFF features two key improvements: learning the Fourier feature basis $B$ and concatenating the input $x$ to the Fourier features. We perform an ablation on several DMControl environments with SAC in Figure 10 to investigate the impact of these modifications.

We first find that training the Fourier feature basis $B$ is critical. Across all four environments, learning is impacted by using a fixed, randomly initialized $B$. This is because finding the right Fourier features at initialization is unlikely in our high dimensional RL problems. Training $B$ allows the network to discover the relevant Fourier features on its own. The relationship with dimension is clear: as the input dimension increases, the performance gap between LFF and fixed Fourier features grows.

Concatenating the input $x$ to the Fourier features is also important. It maintains all of the information that was present in $x$, which is critical in very high dimensional environments. If the Fourier basis $B$ is poorly initialized and it blends together or omits important dimensions of $x$, the network takes a long time to disentangle them, if at all. This problem becomes more likely as the observation dimension increases. While LLF can learn without concatenation in low-dimensional environments like Walker and Finger, it has a much harder time learning in Quadruped and Humanoid.

Finally, we test an alternative approach to initializing the values of $B$. $B$, which has shape $(k \cdot d_{\text{input}}) \times d_{\text{input}}$, is now initialized as $B = (I, cI, c^2 I, \ldots, c^{k-1} I)^\top$ where $I$ is the identity matrix, $k$ is an integer, and $0 < c < 1$ is a tuned multiplier. This parallels the axis-aligned, log-uniform spacing used in NeRF's positional encoding [28], but we additionally concatenate $x$ and train $B$. We find that this initialization method, dubbed "Log-uniform FF" in Figure 10, is consistently worse than sampling from $\mathcal{N}(0, \sigma^2)$. This is likely because the initialization fails to capture features that are not axis-aligned, so most of the training time is used to discover the right combinations of input features.

## 7 Conclusions and Future Work

We highlight that the standard MLP or CNN architecture in state-based and image-based deep RL methods remain susceptible to noise in the Bellman update. To overcome this, we proposed embedding the input using *learned* Fourier features. We show both theoretically and empirically that this encoding enables fine-grained control over the network's frequency-specific learning rate. Our LFF architecture serves as a plug-and-play addition to any state-of-the-art method and leads to consistent improvement in sample efficiency on standard state-space and image-space environments.

One shortcoming of frequency-based regularization is that it does not help when the noise and the signal look similar in frequency space. Future work should examine when this is the case, and test whether other regularization methods are complementary to LFF. Another line of work, partially explored in Appendix F, is using LFF with large $\sigma$ to fit high frequencies and reduce underfitting in model-based or tabular reinforcement learning scenarios. We hope this work will provide new perspectives on existing RL algorithms for the community to build upon.

