**Acknowledgments** We thank Shikhar Bahl, Murtaza Dalal, Wenlong Huang, and Aravind Sivaku-mar for comments on early drafts of this paper. The work was supported in part by NSF IIS-2024594 and GoodAI Research Award. AL is supported by the NSF GRFP. This material is based upon work supported by the National Science Foundation Graduate Research Fellowship Program under Grant No. DGE1745016 and DGE2140739. Any opinions, findings, and conclusions or recommendations expressed in this material are those of the author(s) and do not necessarily reflect the views of the National Science Foundation

# References

[1] J. Achiam, E. Knight, and P. Abbeel. Towards characterizing divergence in deep q-learning. *arXiv preprint arXiv:1903.08894*, 2019. 17, 18

[2] Z. Allen-Zhu and Y. Li. What can resnet learn efficiently, going beyond kernels? *arXiv preprint arXiv:1905.10337*, 2019. 4

[3] B. Bamieh. Discovering transforms: A tutorial on circulant matrices, circular convolution, and the discrete fourier transform. *arXiv preprint arXiv:1805.05533*, 2018. 15

[4] R. Basri, D. Jacobs, Y. Kasten, and S. Kritchman. The convergence rate of neural networks for learned functions of different frequencies. *arXiv preprint arXiv:1906.00425*, 2019. 4, 5, 23

[5] R. Basri, M. Galun, A. Geifman, D. Jacobs, Y. Kasten, and S. Kritchman. Frequency bias in neural networks for input of non-uniform density. In *International Conference on Machine Learning*, pages 685–694. PMLR, 2020. 4

[6] R. Bellman. Dynamic programming. *Science*, 153(3731):34–37, 1966. 2

[7] G. Brockman, V. Cheung, L. Pettersson, J. Schneider, J. Schulman, J. Tang, and W. Zaremba. Openai gym, 2016. 19

[8] L. Chizat, E. Oyallon, and F. Bach. On lazy training in differentiable programming. *arXiv preprint arXiv:1812.07956*, 2018. 4

[9] F. Farnia, J. Zhang, and D. Tse. A spectral approach to generalization and optimization in neural networks. 2018. 23

[10] S. Fort, G. K. Dziugaite, M. Paul, S. Kharaghani, D. M. Roy, and S. Ganguli. Deep learning versus kernel learning: an empirical study of loss landscape geometry and the time evolution of the neural tangent kernel. *arXiv preprint arXiv:2010.15110*, 2020. 4

[11] J. Fu, A. Kumar, M. Soh, and S. Levine. Diagnosing bottlenecks in deep q-learning algorithms. In *International Conference on Machine Learning*, pages 2021–2030. PMLR, 2019. 23

[12] S. Fujimoto, H. Hoof, and D. Meger. Addressing function approximation error in actor-critic methods. In *International Conference on Machine Learning*, pages 1587–1596. PMLR, 2018. 2

[13] F. Gogianu, T. Berariu, M. Rosca, C. Clopath, L. Busoniu, and R. Pascanu. Spectral nor-malisation for deep reinforcement learning: an optimisation perspective. *arXiv preprint arXiv:2105.05246*, 2021. 8

[14] T. Haarnoja, A. Zhou, P. Abbeel, and S. Levine. Soft actor-critic: Off-policy maximum entropy deep reinforcement learning with a stochastic actor. In *International Conference on Machine Learning*, pages 1861–1870. PMLR, 2018. 7, 26, 27

[15] T. Haarnoja, A. Zhou, K. Hartikainen, G. Tucker, S. Ha, J. Tan, V. Kumar, H. Zhu, A. Gupta, P. Abbeel, et al. Soft actor-critic algorithms and applications. *arXiv preprint arXiv:1812.05905*, 2018. 26

[16] H. Hasselt. Double q-learning. *Advances in neural information processing systems*, 23:2613–2621, 2010. 1, 2

[17] A. E. Hoerl and R. W. Kennard. Ridge regression: Biased estimation for nonorthogonal problems. *Technometrics*, 12(1):55–67, 1970. 1

[18] A. Jacot, F. Gabriel, and C. Hongler. Neural tangent kernel: Convergence and generalization in neural networks. *arXiv preprint arXiv:1806.07572*, 2018. 2, 4

[19] D. P. Kingma and J. Ba. Adam: A method for stochastic optimization. *arXiv preprint arXiv:1412.6980*, 2014. 26, 27

[20] J. Z. Kolter, A. Y. Ng, et al. Learning omnidirectional path following using dimensionality reduction. In *Robotics: Science and Systems*, pages 27–30, 2007. 3

[21] G. Konidaris, S. Osentoski, and P. Thomas. Value function approximation in reinforcement learning using the fourier basis. In *Proceedings of the AAAI Conference on Artificial Intelligence*, volume 25, 2011. 3

[22] A. Krogh and J. A. Hertz. A simple weight decay can improve generalization. In *Advances in neural information processing systems*, pages 950–957, 1992. 1

[23] A. Kumar, R. Agarwal, D. Ghosh, and S. Levine. Implicit under-parameterization inhibits data-efficient deep reinforcement learning. *arXiv preprint arXiv:2010.14498*, 2020. 21, 22

[24] M. Laskin, K. Lee, A. Stooke, L. Pinto, P. Abbeel, and A. Srinivas. Reinforcement learning with augmented data. *arXiv preprint arXiv:2004.14990*, 2020. 7, 27

[25] T. P. Lillicrap, J. J. Hunt, A. Pritzel, N. Heess, T. Erez, Y. Tassa, D. Silver, and D. Wierstra. Continuous control with deep reinforcement learning. *arXiv preprint arXiv:1509.02971*, 2015. 9

[26] Z. Liu, X. Li, B. Kang, and T. Darrell. Regularization matters in policy optimization–an empirical study on continuous control. *arXiv preprint arXiv:1910.09191*, 2019. 2

[27] A. McCallum. Reinforcement learning with selective perception and hidden state [ph. d. thesis]. *Computer Science Department, University of Rochester*, 1996. 23

[28] B. Mildenhall, P. P. Srinivasan, M. Tancik, J. T. Barron, R. Ramamoorthi, and R. Ng. Nerf: Representing scenes as neural radiance fields for view synthesis. In *European Conference on Computer Vision*, pages 405–421. Springer, 2020. 3, 10

[29] V. Mnih, K. Kavukcuoglu, D. Silver, A. Graves, I. Antonoglou, D. Wierstra, and M. Riedmiller. Playing atari with deep reinforcement learning. *arXiv preprint arXiv:1312.5602*, 2013. 9

[30] R. Novak, L. Xiao, J. Hron, J. Lee, A. A. Alemi, J. Sohl-Dickstein, and S. S. Schoenholz. Neural tangents: Fast and easy infinite neural networks in python. *arXiv preprint arXiv:1912.02803*, 2019. 6

[31] A. Piché, J. Marino, G. M. Marconi, C. Pal, and M. E. Khan. Beyond target networks: Improving deep $q$-learning with functional regularization. *arXiv preprint arXiv:2106.02613*, 2021. 8

[32] D. Precup, R. S. Sutton, and S. Dasgupta. Off-policy temporal-difference learning with function approximation. In *ICML*, pages 417–424, 2001. 2

[33] N. Rahaman, D. Arpit, A. Baratin, F. Draxler, M. Lin, F. A. Hamprecht, Y. Bengio, and A. C. Courville. On the spectral bias of deep neural networks. 2018. 23

[34] A. Rahimi, B. Recht, et al. Random features for large-scale kernel machines. In *NIPS*, volume 3, page 5. Citeseer, 2007. 2, 3, 4

[35] J. Schulman, P. Moritz, S. Levine, M. Jordan, and P. Abbeel. High-dimensional continuous control using generalized advantage estimation. *arXiv preprint arXiv:1506.02438*, 2015. 19

[36] J. Schulman, F. Wolski, P. Dhariwal, A. Radford, and O. Klimov. Proximal policy optimization algorithms. *arXiv preprint arXiv:1707.06347*, 2017. 19, 27

[37] L. Shao, Y. You, M. Yan, Q. Sun, and J. Bohg. Grac: Self-guided and self-regularized actor-critic. *arXiv preprint arXiv:2009.08973*, 2020. 9

[38] S. Sinha, H. Bharadhwaj, A. Srinivas, and A. Garg. D2rl: Deep dense architectures in reinforcement learning. *arXiv preprint arXiv:2010.09163*, 2020. 4

[39] A. Srinivas, M. Laskin, and P. Abbeel. Curl: Contrastive unsupervised representations for reinforcement learning. *arXiv preprint arXiv:2004.04136*, 2020. 7

[40] N. Srivastava, G. Hinton, A. Krizhevsky, I. Sutskever, and R. Salakhutdinov. Dropout: a simple way to prevent neural networks from overfitting. *The journal of machine learning research*, 15 (1):1929–1958, 2014. 1, 8

[41] R. S. Sutton and A. G. Barto. *Reinforcement learning: An introduction*. MIT press, 2018. 1, 2

[42] M. Tancik, P. P. Srinivasan, B. Mildenhall, S. Fridovich-Keil, N. Raghavan, U. Singhal, R. Ramamoorthi, J. T. Barron, and R. Ng. Fourier features let networks learn high frequency functions in low dimensional domains. *arXiv preprint arXiv:2006.10739*, 2020. 2, 3, 4, 6

[43] Y. Tassa, Y. Doron, A. Muldal, T. Erez, Y. Li, D. d. L. Casas, D. Budden, A. Abdolmaleki, J. Merel, A. Lefrancq, et al. Deepmind control suite. *arXiv preprint arXiv:1801.00690*, 2018. 2, 7

[44] S. Thrun and A. Schwartz. Issues in using function approximation for reinforcement learning. In *Proceedings of the Fourth Connectionist Models Summer School*, pages 255–263. Hillsdale, NJ, 1993. 1

[45] R. Tibshirani. Regression shrinkage and selection via the lasso. *Journal of the Royal Statistical Society: Series B (Methodological)*, 58(1):267–288, 1996. 1

[46] J. N. Tsitsiklis and B. Van Roy. An analysis of temporal-difference learning with function approximation. *IEEE transactions on automatic control*, 42(5):674–690, 1997. 1

[47] H. van Hasselt, A. Guez, and D. Silver. Deep reinforcement learning with double q-learning, 2015. 1, 2

[48] H. Van Hasselt, Y. Doron, F. Strub, M. Hessel, N. Sonnerat, and J. Modayil. Deep reinforcement learning and the deadly triad. *arXiv preprint arXiv:1812.02648*, 2018. 3

[49] Z.-Q. J. Xu, Y. Zhang, T. Luo, Y. Xiao, and Z. Ma. Frequency principle: Fourier analysis sheds light on deep neural networks. *arXiv preprint arXiv:1901.06523*, 2019. 23

[50] Y. Yao, L. Rosasco, and A. Caponnetto. On early stopping in gradient descent learning. *Constructive Approximation*, 26(2):289–315, 2007. 1

[51] D. Yarats and I. Kostrikov. Soft actor-critic (sac) implementation in pytorch. https://github.com/denisyarats/pytorch_sac, 2020. 26


# A  Theoretical Analysis

## A.1  Functional Convergence Rate

We provide Lemma 2 and a quick proof sketch as background for readers who are not familiar with the neural tangent kernel literature.

**Lemma 2.** *When training an infinite width neural network via gradient flow on the squared error, the training residual at time $t$ is:*

$$f_{\theta_t}(x) - y = e^{-\eta K t}(f_{\theta_0}(x) - y) \tag{11}$$

*where $f_{\theta_t}(x)$ is the column vector of model predictions for all $x_i$, $y$ is the column vector of stacked training labels, $\eta$ is the multiplier for gradient flow, and $K$ is the NTK kernel matrix with $K_{ij} = \langle \nabla_\theta f_{\theta_0}(x_i), \nabla_\theta f_{\theta_0}(x_j) \rangle$.*

*Proof.* The squared error $L(\theta, x) = \|f_\theta(x) - y\|_2^2$ has gradient:

$$\nabla_\theta L(\theta, x) = \nabla_\theta f_\theta(x)^\top (f_\theta(x) - y) \tag{12}$$

Since we train with gradient flow, the parameters change at rate:

$$\frac{d\theta_t}{dt} = -\eta \nabla_{\theta_t} L(\theta_t, x) \tag{13}$$

$$= -\eta \nabla_{\theta_t} f_{\theta_t}(x)^\top (f_{\theta_t}(x) - y) \tag{14}$$

By the chain rule,

$$\frac{df_{\theta_t}(x)}{dt} = \frac{df_{\theta_t}(x)}{d\theta_t} \frac{d\theta_t}{dt} \tag{15}$$

$$= -\eta \nabla_{\theta_t} f_{\theta_t}(x) \nabla_{\theta_t} f_{\theta_t}(x)^\top (f_{\theta_t}(x) - y) \tag{16}$$

$$= -\eta K_{\theta_t}(f_{\theta_t}(x) - y) \tag{17}$$

where $K$ is the NTK kernel matrix at time $t$, with entries $K_{ij} = \langle \nabla_{\theta_t} f_{\theta_t}(x_i), \nabla_{\theta_t} f_{\theta_t}(x_j) \rangle$. Since $y$ is a constant, and $K_{\theta_t} \approx K_{\theta_0} \triangleq K$ due to the infinite-width limit, we can write this as:

$$\frac{d(f_{\theta_t}(x) - y)}{dt} = -\eta K(f_{\theta_t}(x) - y) \tag{18}$$

This is a well-known differential equation, with closed form solution:

$$f_{\theta_t}(x) - y = e^{-\eta K t}(f_{\theta_0}(x) - y) \tag{19}$$

$$\square$$

## A.2  Eigenvalues of the NTK matrix and the Discrete Fourier Transform

**Lemma 3.** *A circulant matrix $C \in \mathbb{R}^{n \times n}$ with first row $(c_0, \ldots, c_{n-1})$ has eigenvectors $\{x^{(k)}\}_{k=1}^n$ corresponding to the column vectors of the DFT matrix:*

$$x^{(k)} = (\omega_n^{0k}, \omega_n^{1k}, \ldots, \omega_n^{(n-1)k})^\top \tag{20}$$

*where $\omega_n = e^{\frac{2\pi i}{n}}$ is the $n$th root of unity. The corresponding eigenvalue is the $k$th DFT value $\lambda^{(k)} = DFT(c_0, \ldots, c_{n-1})_k$.*

*Proof.* This is a well-known property of circulant matrices [3]. Nevertheless, we provide a simple proof here. First, let's make clear the structure of the circulant matrix $C$:

$$C = \begin{bmatrix} c_0 & c_1 & c_2 & \ldots & c_{n-1} \\ c_{n-1} & c_0 & c_1 & \ldots & c_{n-2} \\ c_{n-2} & c_{n-1} & c_0 & \ldots & c_{n-3} \\ \vdots & \vdots & \vdots & \ddots & \vdots \\ c_1 & c_2 & c_3 & \ldots & c_0 \end{bmatrix} \tag{21}$$

Again, we want to show that an eigenvector of $C$ is $x^{(k)}$, the $k$th column of the DFT matrix $F$.

$$x^{(k)} = (\omega_n^{0k}, \omega_n^{1k}, \ldots, \omega_n^{(n-1)k})^\top \tag{22}$$

where $\omega_n = e^{\frac{2\pi i}{n}}$ is the $n$th root of unity. Note that for all positive integers $k$, $\omega_n^{jk}$ is periodic in $j$ with period $n$. This is because:

$$\omega_n^{nk} = e^{\frac{2\pi nki}{n}} \tag{23}$$
$$= \cos(2\pi k) + i\sin(2\pi k) \tag{24}$$
$$= 1 \tag{25}$$

Now, let us show that $x^{(k)}$ is an eigenvector of $C$. Let $y = Cx^{(k)}$. The $i$th element of $y$ is then

$$y_i = \sum_{j=0}^{n-1} c_{j-i} \omega_n^{jk} \tag{26}$$

$$= \omega_n^{ik} \sum_{j=0}^{n-1} c_{j-i} \omega_n^{(j-i)k} \tag{27}$$

The remaining sum does not depend on $i$, since $c_{j-i}$ and $\omega_n^{(j-i)k}$ are periodic with period $n$. This means we can rearrange the indices of the sum to get:

$$y_i = \omega_n^{ik} \sum_{j=0}^{n-1} c_j \omega_n^{jk} \tag{28}$$

$$= x_i^{(k)} \lambda_k \tag{29}$$

where $\lambda_k = \sum_{j=0}^{n-1} c_j \omega_n^{jk}$ is exactly the $k$th term in the DFT of the signal $(c_0, c_1, \ldots, c_{n-1})$. Thus, $Cx^{(k)} = \lambda_k x^{(k)}$, so $x^{(k)}$ is an eigenvector of $C$ with corresponding eigenvalue $\lambda^{(k)} = DFT(c_0, \ldots, c_{n-1})_k$. $\qquad\square$

### A.3 Proof of Lemma 1: NTK for 2 layer Fourier feature model

*Proof.* The gradient consists of two parts: the gradient with respect to $B$ and the gradient with respect to $W$. We can calculate the NTK for each part respectively and then sum them.

**For $W$:**

$$\nabla_W f(x) = \sqrt{\frac{2}{m}} \begin{bmatrix} \sin(Bx) \\ \cos(Bx) \end{bmatrix} \tag{30}$$

The width-$m$ kernel is then:

$$k_m^W(x, x') = \frac{2}{m} \sum_{i=1}^{m/2} \cos(b_i^\top x)\cos(b_i^\top x') + \sin(b_i^\top x)\sin(b_i^\top x') \tag{31}$$

Using the angle difference formula, this reduces to:

$$k_m^W(x, x') = \frac{2}{m} \sum_{i=1}^{m/2} \cos(b_i^\top(x - x')) \tag{32}$$

As $m \to \infty$, this converges to a deterministic kernel $k^W(x, x') = \mathbb{E}_{B_{ij} \sim \mathcal{N}(0,\tau)}[\cos(b_i^\top(x - x'))]$.

Using the fact that $E_{X \sim \mathcal{N}(0,\Sigma)}[\cos(t^\top X)] = \exp\{-\frac{1}{2}t^\top \Sigma t\}$ for fixed vector $t$ and the fact that $\Sigma = \text{diag}(\sigma^2, \sigma^2)$ in our case, this kernel function simplifies to:

$$k^W(x, x') = \exp\left\{-\frac{\sigma^2}{2}\|x - x'\|_2^2\right\} \tag{33}$$

**For $B$:**

$$\nabla_{b_i} f(x) = \sqrt{\frac{2}{m}} \left( W_i \cos(b_i^\top x) - W_{i+m/2} \sin(b_i^\top x) \right) x \tag{34}$$

The width-$m$ kernel for $B$ is then:

$$k_m^B(x, x') = \frac{2x^\top x'}{m} \sum_{i=1}^{m/2} W_i^2 \left( \cos(b_i^\top x) \cos(b_i^\top x') \right) + W_{i+m/2}^2 \left( \sin(b_i^\top x) \sin(b_i^\top x') \right) \tag{35}$$
$$- W_i W_{i+m/2} \left( \cos(b_i^\top x) \sin(b_i^\top x') + \sin(b_i^\top x) \cos(b_i^\top x') \right)$$

As we take $m \to \infty$, recall that $W_j \sim \mathcal{N}(0, 1)$ i.i.d.. Thus, $\mathbb{E}[W_j^2] = 1$ and $\mathbb{E}[W_j W_k] = 0$ for $j \neq k$. The NTK is then

$$k^B(x, x') = x^\top x' \mathbb{E} \left[ \cos(b_i^\top x) \cos(b_i^\top x') + \sin(b_i^\top x) \sin(b_i^\top x') \right] \tag{36}$$

The interior of the expectation is exactly the same as the summand in Equation 31. Following the same steps, we get the simplified kernel function:

$$k^B(x, x') = x^\top x' \exp \left\{ -\frac{\sigma^2}{2} \|x - x'\|_2^2 \right\} \tag{37}$$

Finally, our overall kernel function $k(x, x') = k^W(x, x') + k^B(x, x')$ is:

$$k(x, x') = \left( 1 + x^\top x' \right) \exp \left\{ -\frac{\sigma^2}{2} \|x - x'\|_2^2 \right\} \tag{38}$$

Since $x, x' \in \mathbb{S}^{d-1}$, they have unit norm, with $\|x - x'\|_2^2 = 2(1 - x^\top x') = 2(1 - \cos\theta)$. This gives us two equivalent forms of the NTK:

$$k(x, x') = \left( 2 - \frac{\|x - x'\|_2^2}{2} \right) \exp \left\{ -\frac{\sigma^2}{2} \|x - x'\|_2^2 \right\} \tag{39}$$
$$= (1 + \cos\theta) \exp \left\{ \sigma^2 (\cos\theta - 1) \right\} \tag{40}$$

$\square$

## A.4   When is the Bellman Update a Contraction?

Here, we examine LFF's stability under Bellman updates by using results from Achiam et al. [1], who used the NTK approximation to prove that the Bellman update is a contraction in finite MDPs if

$$\forall i, \quad \alpha K_{ii} \rho_i < 1 \tag{41}$$
$$\forall i, \quad (1 + \gamma) \sum_{j \neq i} |K_{ij}| \rho_j \leq (1 - \gamma) K_{ii} \rho_i \tag{42}$$

where $K$ is the NTK of the Q-network and $\rho_i$ is the density of transition $i$ in the replay buffer. Intuitively, $K_{ij}$ measures the amount that a gradient update on transition $j$ affects the function output on transition $i$. In order for the Bellman update to be a contraction, the change at $(s_i, a_i)$ due to gradient contributions from all other transitions should be relatively small. This suggests that LFF, with very large $\sigma$, could fulfill the conditions in Equation 43 and 44. We formalize this in Theorem 1:

**Theorem 1.** *For a finite MDP with the state-action space as a finite, uniform subset of $\mathcal{S}^d$, and when we have uniform support for each transition in the replay buffer, the Bellman update on a 2 layer LFF architecture is a contraction for suitably small learning rate $\alpha$.*

*Proof.* We need our kernel to satisfy two conditions [1]:

$$\forall i, \quad \alpha K_{ii} \rho_i < 1 \tag{43}$$
$$\forall i, \quad (1 + \gamma) \sum_{j \neq i} |K_{ij}| \rho_j \leq (1 - \gamma) K_{ii} \rho_i \tag{44}$$

Equation 43 is easy to satisfy, as we assumed that all transitions appear uniformly in our buffer, and we know from Lemma 1 that $k(x, x) = 2$. Thus, we only need to make the learning rate $\alpha$ small enough such that $\alpha < \frac{1}{2\rho_i}$.

For Equation 44, we can prove a loose lower bound on the variance $\sigma^2$ that is required for the Bellman update to be a contraction. As stated in the main text, we assume that we have $N + 1$ datapoints $x_i$ that are distributed approximately uniformly over $\mathbb{S}^{d-1}$. Here, uniformly simply implies that we have an upper bound $x_i^\top x_j = \cos\theta < 1 - \delta$ for all $i \neq j$ and fixed positive $\delta > 0$.

First, we bound $|K_{ij}|$ for all $i \neq j$. Using the expression from Lemma 1 and our upper bound on $\cos\theta$, we have $|K_{ij}| \leq (2 - \delta) \exp\left\{-\delta\sigma^2\right\}, \forall i \neq j$. Then, plugging this into Equation 44 and cancelling the buffer frequencies $\rho_i$, which are equal by assumption,

$$N(1 + \gamma)(2 - \delta) \exp\left\{-\delta\sigma^2\right\} \leq 2(1 - \gamma) \tag{45}$$

Rearranging gives us:

$$\sigma^2 \geq \frac{1}{\delta} \log \frac{N(1 + \gamma)(2 - \delta)}{2(1 - \gamma)} \tag{46}$$

As long as $\alpha$ is small enough and $\gamma < 1$, an infinite-width LFF architecture initialized with a suitably large $\tau$ will enjoy Bellman updates that are always contractions in the sup-norm. $\qquad\square$

**Note** However, this does not align with what we see in practice, where small $\sigma$ yields the best performance, and increasing $\sigma$ leads to worse performance. This is because Achiam et al. [1] makes the assumption that the replay buffer assigns positive probability to every possible transition, which is impossible in our continuous MDPs. We also do stochastic optimization with minibatches, which further deviates from the theory. Finally, guarantee of a contraction does not imply sample efficiency. Indeed, an algorithm that contracts slowly at every step can be much worse than an algorithm that greatly improves in expectation.

# B  On-policy Results

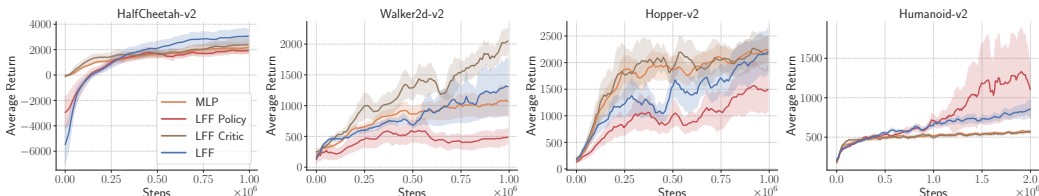

Figure 11: **On-Policy Evaluation**: We train PPO on 4 OpenAI Gym environments with our LFF architecture and a vanilla MLP, both of which have roughly the same number of parameters. LFF does not produce a consistent gain in sample efficiency here, which is consistent with our hypothesis that LFF helps only for off-policy RL.

## B.1  On-policy LFF Setup

We evaluate proximal policy optimization (PPO, Schulman et al. [36]) on 4 environments from OpenAI gym [7]. These environments range from easy, e.g. HalfCheetah-v2, to difficult, e.g. Humanoid-v2. Just as we did in the off-policy setup, we modify only the architecture. keeping the hyperparameters fixed. We compare MLPs with 3 hidden layers to LFF with our Fourier feature input layer followed by 2 hidden layers. We use $d_{\text{fourier}} = 1024$ and $\sigma = 0.001$ for all environments and also test what happens if we use LFF for only the policy or only the critic.

## B.2  LFF architecture for on-policy RL

In Figure 11, we show the results of using LFF for PPO. Unlike LFF on SAC, LFF did not yield consistent gains for PPO. The best setting was to use LFF for only the critic, which does as well as MLPs on 3 environments (HalfCheetah, Hopper, and Humanoid) and better on Walker2d, but this is only a modest improvement. This is not surprising and is likely because policy gradient methods have different optimization challenges than those based on Q-learning. For one, the accuracy of the value function baseline is less important for policy gradient methods. The on-policy cumulative return provides a lot of reward signal, and the value function baseline mainly reduces variance in the gradient update. In addition, generalized advantage estimation [35] further reduces variance. Thus, noise in the bootstrapping process is not as serious of a problem as in off-policy learning.

# C  Performance Under Added Noise

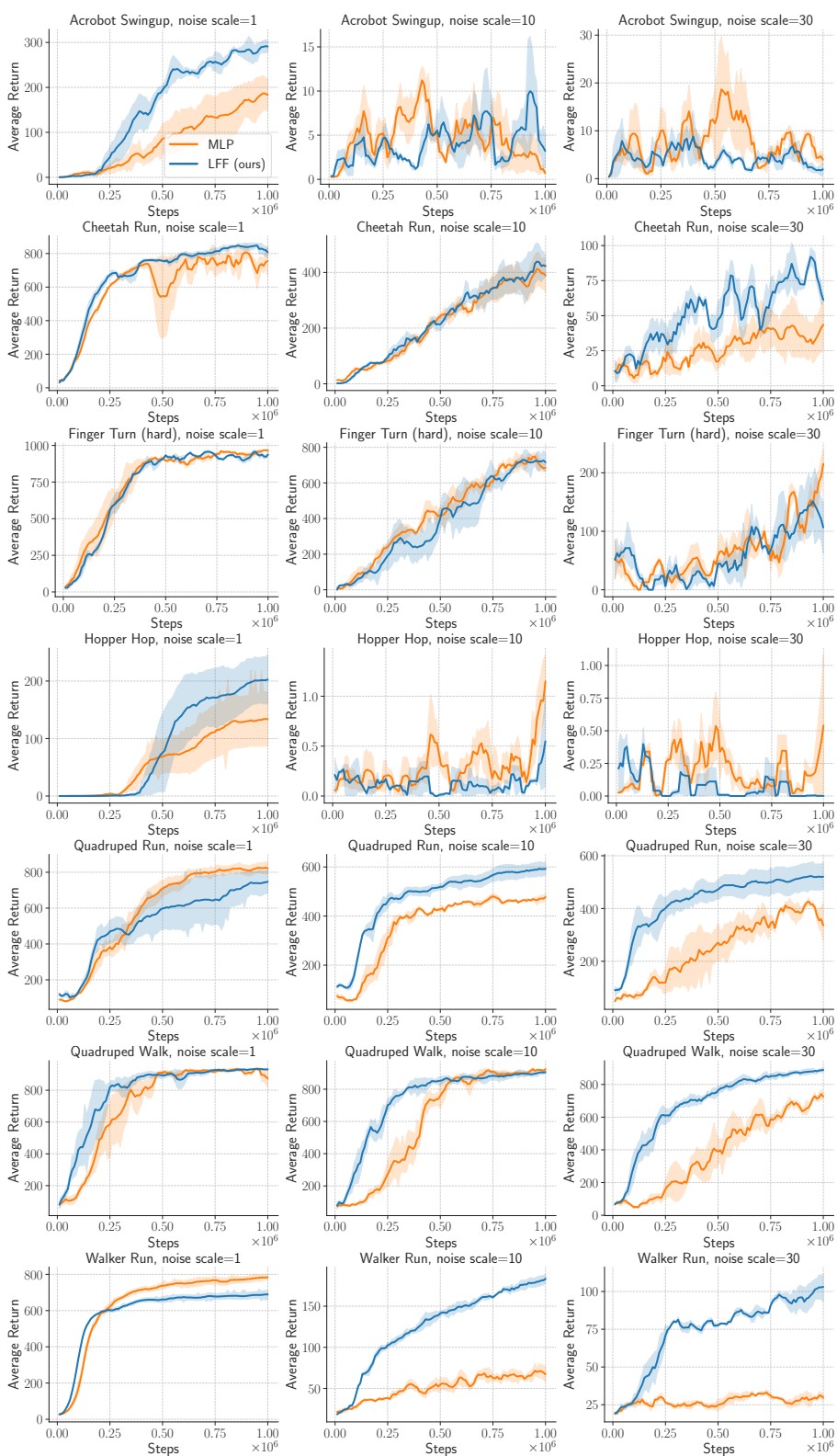

Figure 12: **State-based with Added Noise**: We add zero-mean Gaussian noise to the targets. As the standard deviation of the added noise increases, LFF maintains its performance better than MLPs.

# D   Noise Amplification vs Implicit Underparameterization in Gridworld MDP

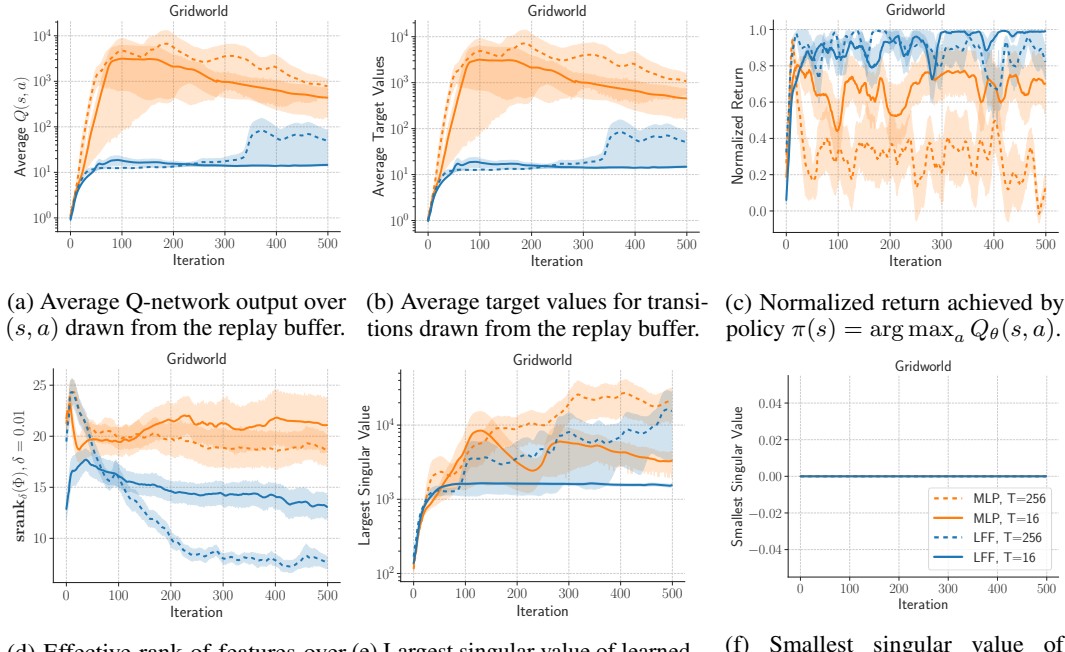

(a) Average Q-network output over $(s, a)$ drawn from the replay buffer.

(b) Average target values for transitions drawn from the replay buffer.

(c) Normalized return achieved by policy $\pi(s) = \arg\max_a Q_\theta(s, a)$.

(d) Effective rank of features over minibatch of states.

(e) Largest singular value of learned features.

(f) Smallest singular value of learned features.

Figure 13: Measuring noise amplification vs implicit underparameterization on GRID16SMOOTHOBS.

We want to examine whether noise amplification is indeed a problem that off-policy deep RL methods suffer from. Furthermore, if it is happening, Kumar et al. [23] hypothesize that it is being caused by underfitting ("implicit underparameterization"), which would contradict our claim that LFF improves learning by regularizing the training dynamics. We test this in Figure 13 by performing Fitted Q-iteration (FQI) on the GRID16SMOOTHOBS environment from Kumar et al. [23]. GRID16SMOOTHOBS is a discrete environment with 256 states and 5 actions, so we can use Q-iteration to calculate the optimal $Q^*$ and compare that with our learned $Q_\theta$. $T$ denotes the number of gradient steps per FQI iteration; increasing the number of gradient steps is empirically more likely to cause divergence for MLP Q-functions. Note that we use a log scale for the y-axis in (a,b,e).

**Noise amplification**   **(a)** shows that MLP-based Q-functions steadily blow up to orders of magnitude above the true $Q^*$, whose average value is around 15 in this environment. In contrast, LFF-based Q-functions either converge stably to the correct magnitude, or resist increasing as much. **(b)** shows that the MLP target values are in a positive feedback loop with the Q-values. **(c)** shows that divergence coincides with a drop in the returns. Together, these results indicate that there can be a harmful feedback loop in the bootstrapping process, and that methods like LFF, which reduce fitting to noise, can help stabilize training.

**Implicit Underparameterization**   We run Fitted Q-iteration and calculate the effective rank of the Q-network, which we parameterize using either a MLP or LFF network. We follow the procedure from Kumar et al. [23]: sample 2048 states from the replay buffer and calculate the singular values $\sigma_i$ of the aggregated feature matrix $\Phi$. The effective rank is then defined as $\mathbf{srank}_\delta(\Phi) = \min\left\{k : \frac{\sum_{i=1}^{k} \sigma_i(\Phi)}{\sum_{j=1}^{d} \sigma_j(\Phi)} \geq 1 - \delta\right\}$. **(d)** shows that the MLP's effective rank does not actually drop over training. Furthermore, LFF is able to avoid diverging $Q$-values, even though it has signficantly lower srank than its MLP counterpart. Thus, noise amplification for MLPs in this setting is likely not related to any underfitting measured by the effective rank. **(e)** shows that the

largest singular value blows up for MLPs, but stabilizes when training LFF for a reasonable number of gradient steps. **(f)** shows that the minimum singular value stays at zero over the course of training. In the context of Kumar et al. [23], this implies that their penalty $\sigma_{max}(\Phi)^2 - \sigma_{min}(\Phi)^2$ is exactly equivalent to penalizing only the maximum singular value $\sigma_{max}(\Phi)^2$ when using gradient descent. This is because the gradient of the second term is zero when the smallest singular value is zero. Thus, Kumar et al. [23]'s penalty works by constraining the largest singular value and regularizing the magnitude of the feature matrix. Overall, these results support our hypothesis that noise amplification is a problem that is not caused by underfitting and can be ameliorated by regularization.

# E  Sensitiviy Analyses

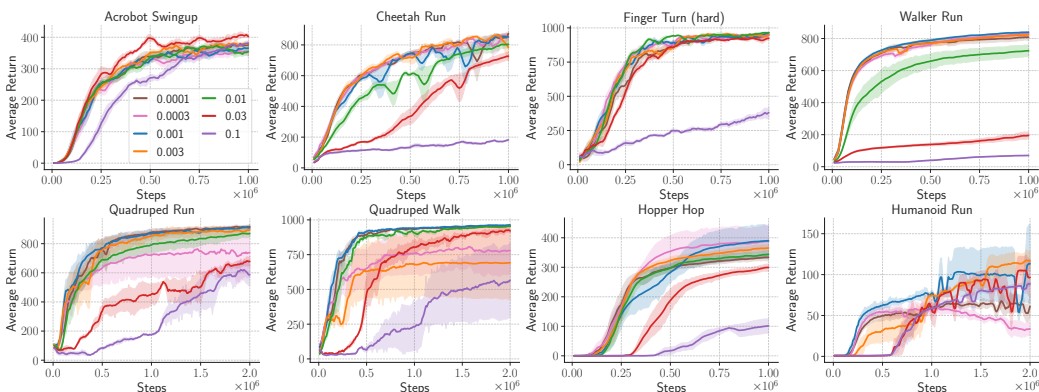

Figure 14: **Sensitivity to $\sigma$.** $\sigma = 0.001$ is a good default across all of these state-based environments. Results are averaged over 5 seeds, using a Fourier dimension of 1024, and the shaded region denotes 1 standard error.

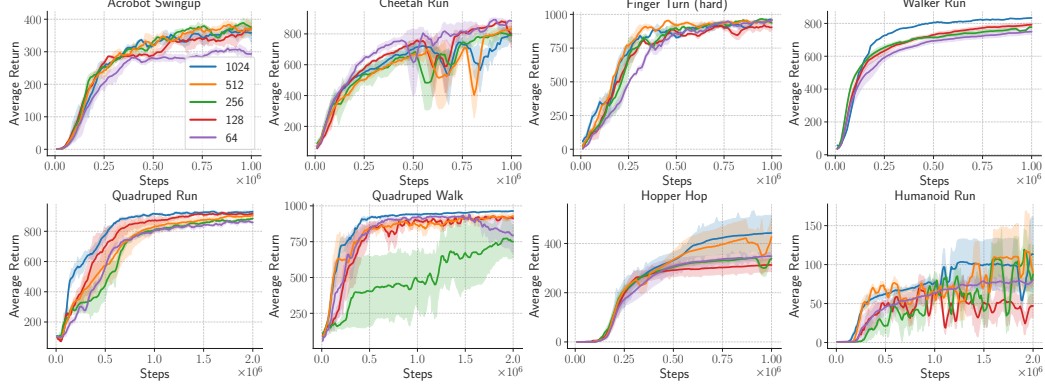

Figure 15: **Sensitivity to the Fourier dimension.** A Fourier dimension of 1024 is a good default across all of these state-based environments. Results are averaged over 5 seeds, using $\sigma = 0.001$, and the shaded region denotes 1 standard error.

# F High Frequency Learning with Learned Fourier Features

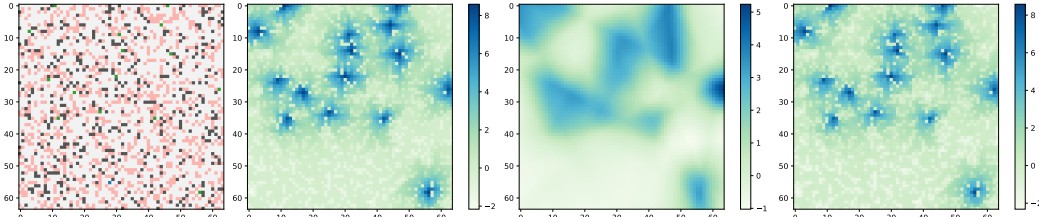

Figure 16: **Potential Underfitting in RL**. Left: gridworld structure. Red squares are lava, green are goals, gray are walls, and white squares are empty. Middle left: ground truth value function $V^*(s)$. Middle right: we fit $Q^*(s, a)$ with an MLP, then display $\max_a Q_\theta(s, a)$. The result is blurry and cannot properly distinguish between critical high and low value states. Right: we fit $Q^*(s, a)$ with our proposed learned Fourier feature architecture, then display $\max_a Q_\theta(s, a)$. It is able to exactly reproduce the ground truth, even with the same number of parameters and gradient steps.

While the main paper focused on using LFF with small $\sigma$ to bias networks towards mainly learning low frequency functions, we can still use larger $\sigma$ to encourage faster high-frequency learning. Prior theoretical work [4, 9, 33, 49] found that ReLU MLPs suffer from *spectral bias* – they can take an impractically long time to fit the highest frequencies. In the reinforcement learning setting, this can cause underfitting when fitting high frequencies is desirable.

## F.1 Gridworld

We demonstrate that underfitting is indeed happening in RL. We create a toy $64 \times 64$ gridworld task [11] in Figure 16, where each square can be one of five types: start state, goal state, empty, wall, or lava. The agent starts at the start state and receives +1 reward for every timestep at a goal state, -1 penalty for every timestep at a lava square, and 0 reward otherwise. It cannot enter cells with a wall. The agent has five actions available: up, down, left, right, or no-op. Whenever the agent takes a step, there is a 20% chance of a random action being taken instead, so it is important for the agent to stay far from lava, lest it accidentally fall in. We use a discount of $\gamma = 0.9$. To create the environment, we randomly initialize it with 25% lava squares and 10% wall squares. We learn the optimal $Q^*$ using Q-iteration, then attempt to fit various neural network architectures with parameters $\theta$ to the ground truth $Q^*$ values through supervised learning:

$$\theta^* = \arg\min_\theta \sum_{s \in \mathcal{S}, a \in \mathcal{A}} (Q_\theta(s, a) - Q^*(s, a))^2 \tag{47}$$

We then try to fit $Q^*$ using a standard MLP with 3 layers and 256 hidden units, and using our proposed architecture with an equivalent number of parameters and $\sigma = 3$. Due to the challenge of visualizing $Q(s, a)$ with 5 actions, we instead show $V(s) = \max_a Q(s, a)$ in Figure 16.

Surprisingly, MLPs have extreme difficulty fitting $Q^*(s, a)$, even when doing supervised learning on this low-dimensional toy example. Even without the challenges of nonstationarity, bootstrapping, and exploration, the deep neural network has trouble learning the optimal Q-function. The learned Q-function blurs together nearby states, making it impossible for the agent to successfully navigate narrow corridors of lava. Prior work has described this problem as state aliasing [27], where an agent conflates separate states in its representation space. We anticipate that this problem is worse in higher dimensional and continuous MDPs and is pervasive throughout reinforcement learning.

In contrast, our LFF embedding with $\sigma = 3$ is able to perfectly learn the ground-truth Q-function. This indicates that LFF with large $\sigma$ can help our Q-networks and policies fit key high-frequency details in certain RL settings. We believe that there are two promising applications for high-frequency learning with LFF: model-based RL and tabular problems. Model-based RL requires modeling the dynamics, which can have sharp changes, such as at contact points. Modeling transition dynamics is also supervised, so there are no bootstrap noise problems exacerbated by accelerating the rate at which high-frequencies are learned. Tabular problems are also suited for high-frequency learning, as they often have sharp changes in dynamics or rewards (e.g. gridworld squares with walls, cliffs, or

lava). Capturing high-frequencies with LFF could improve both the sample-efficiency and asymptotic performance in this setting.

## G Fourier Basis Variance After Training

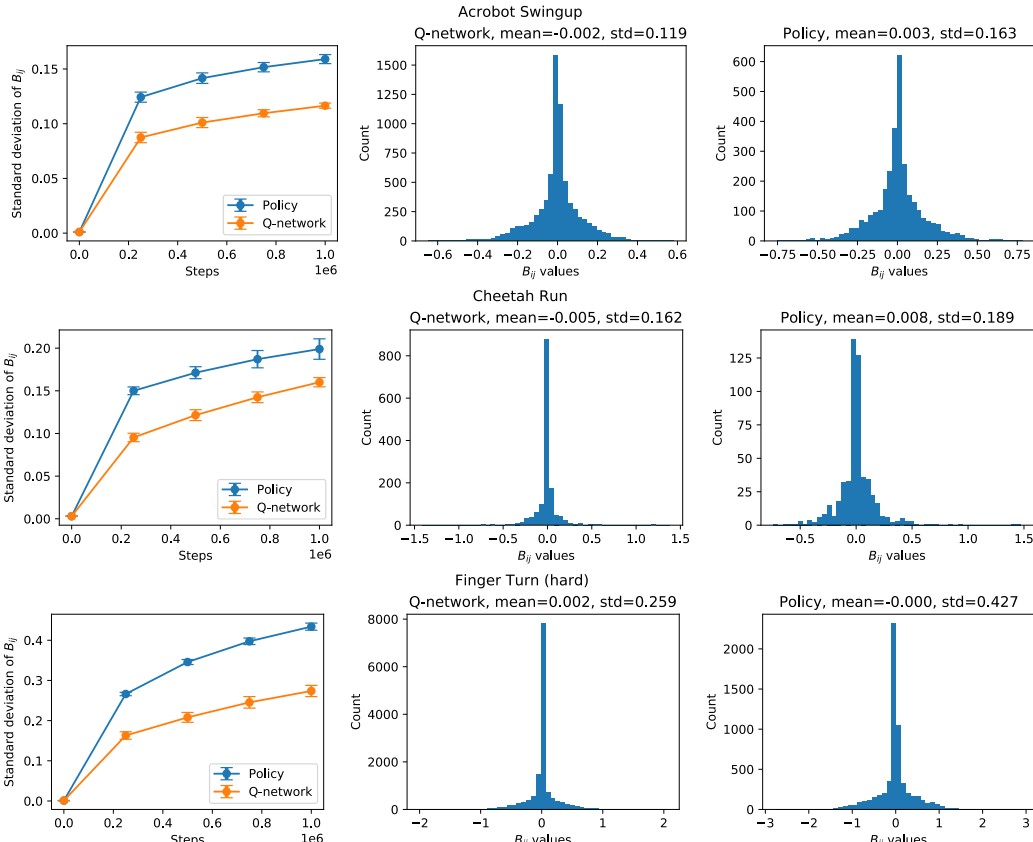

Figure 17: **Change in the standard deviation of Fourier basis entries** $B_{ij}$. The curves on the left show how the standard deviation changes from $\sigma$ at initialization as state-based SAC training progresses. The confidence interval indicates the standard deviation of the standard deviation, measured across 5 seeds. The histograms in the middle and left show the distribution of entries within $B$ for the policy and Q-networks at the end of training.

In Section 4, we used the Neural Tangent Kernel (NTK) perspective to show that the initialization variance of the Fourier basis $B$ controls the per-frequency learning rate in infinite-width neural networks. However, the NTK's infinite-width assumption implies that the entries of the Fourier basis do not change over the course of training. Since we train $B$ with finite width, we examine how its variance evolves over training, which affects its per-frequency learning rate at each point in time. Figure 17 and 18 show how the standard deviation of the Fourier basis change for the policy and Q-network. The standard deviation generally increases from $\sigma = 0.001$ or $\sigma = 0.003$ to about 0.1, which should still be more biased towards low frequencies than vanilla MLPs are (see Figure 4). Furthermore, the increase in standard deviation could actually be desirable. Having more data at the end of training could reduce the impact of bootstrap noise, so there may be less need for smoothing with small $\sigma$. Larger $\sigma$ could bias the network towards fitting medium-frequency signals that are important for achieving full asymptotic efficiency. This could explain the image-based results in Figure 19, where the standard deviation rises to about 0.5.

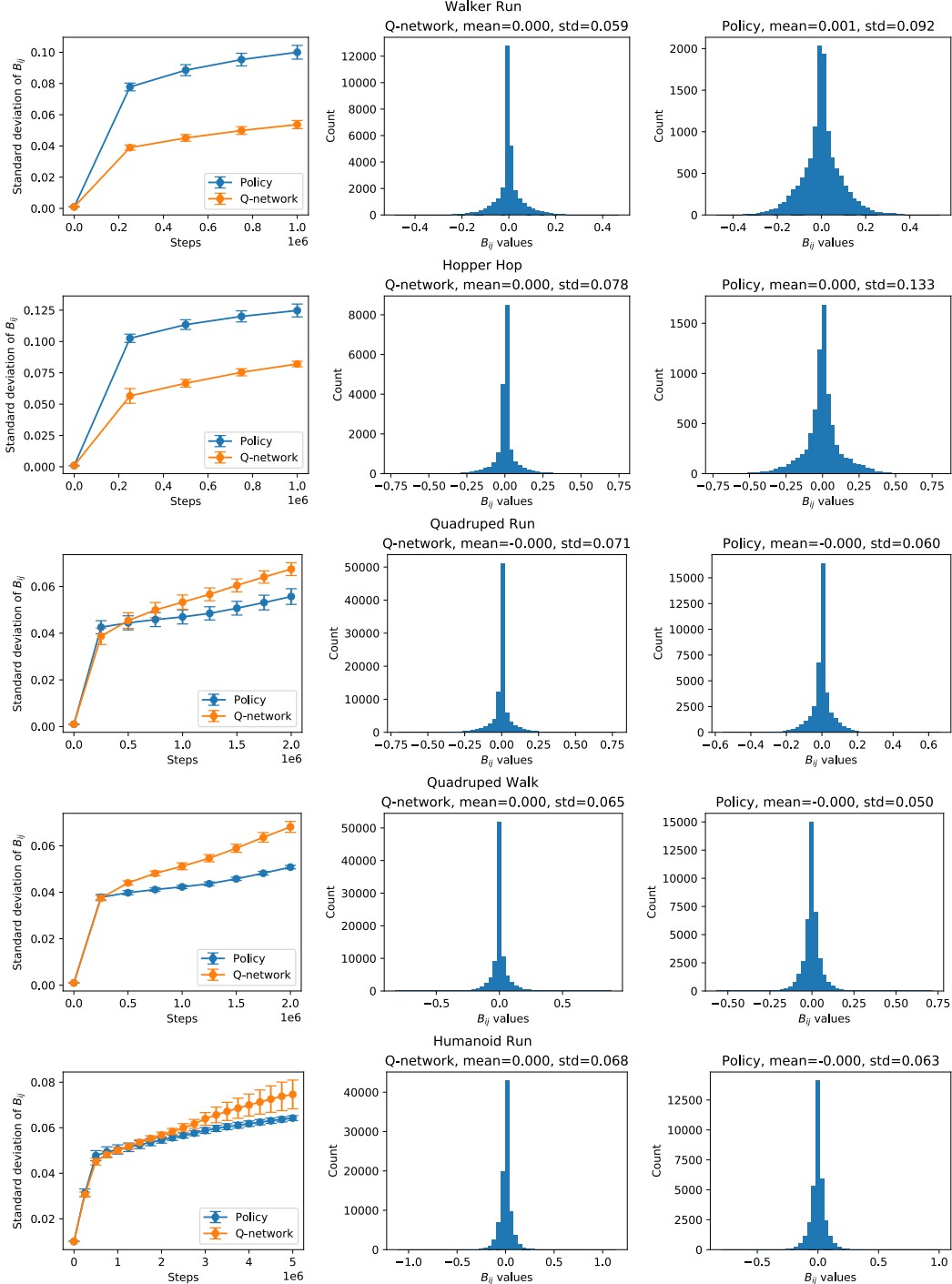

Figure 18: Continuation of Figure 17, which shows how the standard deviation of the Fourier basis changes over training.

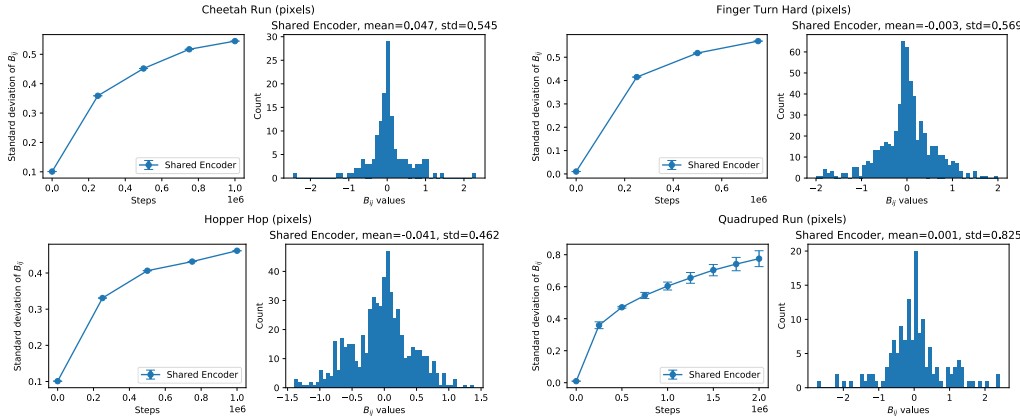

Figure 19: **Change in the standard deviation of Fourier basis entries** $B_{ij}$ **for image-based experiments**. For each of the four environments, we show how the standard deviation of the shared encoder's Fourier basis changes over training. The histogram shows the distribution of the entries of the shared encoder's $B$ at the end of training.

# H   Hyperparameters

We list hyperparameters for state-based and image-based SAC experiments in Table 1 and Table 2, respectively. We also show hyperparameters for PPO experiments from Appendix B in Table 3.

| Parameter | Value |
|---|---|
| Algorithm | Soft Actor Critic [14] |
| Starting codebase | Yarats and Kostrikov [51] |
| Optimizer | Adam [19] |
| Adam $(\beta_1, \beta_2)$ | (0.9, 0.999) |
| Discount | 0.99 |
| Batch size | 1024 |
| Target smoothing coefficient $(\tau)$ | 0.005 |
| Reward scale | Auto-tuned [15] |
| Actor learning rate | $10^{-4}$ |
| Critic learning rate | $10^{-4}$ |
| Reward scale learning rate | $10^{-4}$ |
| Number of exploratory warmup steps | 5000 |
| Number of hidden layers | 3 for MLP, 2 for LFF |
| Hidden size | 1024 |
| Hidden nonlinearity | ReLU |
| Fourier dimension | 64 for Cheetah, 1024 otherwise |
| Standard deviation $\sigma$ of Fourier basis initialization | 0.003 for Cheetah, 0.001 otherwise |

Table 1: Hyperparameters used for the state-basd SAC experiments.

| Parameter | Value |
|---|---|
| Algorithm | Soft Actor Critic [14] + RAD [24] |
| Starting codebase | Laskin et al. [24] |
| Augmentation | Translate: Cheetah. Crop: otherwise. |
| Observation rendering | $(100, 100)$ |
| Observation down/upsampling | Crop: $(84, 84)$. Translate: $(108, 108)$. |
| Replay buffer size | 100000 |
| Initial steps | 1000 |
| Stacked frames | 3 |
| Action repeat | 4 |
| Optimizer | Adam [19] |
| Adam $(\beta_1, \beta_2)$ | $(0.5, 0.999)$ for entropy, $(0.9, 0.999)$ otherwise |
| Learning rate | $10^{-4}$ |
| Batch size | 512 |
| Encoder smoothing coefficient $(\tau)$ | 0.05 |
| Q-network smoothing coefficient $(\tau)$ | 0.01 |
| Critic target update freq | 2 |
| Convolutional layers (excluding LFF embedding) | 4 for LFF, 5 otherwise |
| Discount | 0.99 |
| Fourier dimension | 128 for Hopper, 64 otherwise |
| Initial standard deviation $\sigma$ of Fourier basis | 0.1 for Hopper, Cheetah; 0.01 for Finger, Quadruped |

Table 2: Hyperparameters used for the image-based SAC experiments.

| Parameter | Value |
|---|---|
| Algorithm | Proximal Policy Optimization [36] |
| Learning rate | $3 \times 10^{-4}$ |
| Learning rate decay | linear |
| Entropy coefficient | 0 |
| Value loss coefficient | 0.5 |
| Clip parameter | 0.2 |
| Environment steps per optimization loop | 2048 |
| PPO epochs per optimization loop | 10 |
| Batch size | 64 |
| GAE $\lambda$ | 0.95 |
| Discount | 0.99 |
| Total timesteps | $10^6$ |
| Number of hidden layers | 2 |
| Hidden size | 256 |
| Hidden nonlinearity | $\tanh$ |
| Fourier dimension | 64 |
| Variance of Fourier basis initialization $\tau$ | 0.01 |

Table 3: Hyperparameters used for the PPO experiments.