# OpenReview forum: "Functional Regularization for Reinforcement Learning via Learned Fourier Features"
_NeurIPS.cc/2021/Conference — NeurIPS 2021 Poster_

### Official Review · Reviewer_Susc · 2021-07-08

**Rating:** 7
**Confidence:** 4

**Summary:**

This paper proposes to augment the input space of deep value functions with a finetuned random projection followed by two periodic activations, both cos and sin. An interesting interpretation of this parameterization, Learned Fourier Features (LF2), is that it is an implicit Fourier decomposition of the input space, and that the variance used to initialize the random projection is monotonically related to the dominant frequencies that these random features capture (i.e. $\sigma^2 \to 0$ captures lower frequencies).

It is hypothesized that deep value networks tend to capture high-frequency features, which lead to poor Bellman updates, and thus that LF2 should help by reducing the capture of these high-frequency noise features.
In practice, LF2 does improve performance on some control tasks of the DMControl benchmark.

**Ethical Concerns:**

No ethical concerns.

**Limitations And Societal Impact:**

The authors on superficially address the limitations of their method. More could be done, for example, experiments could show the kinds of environments/observation spaces where their method fails (a failure setup similar to the one in Figure 2 should be easy to create).

There are no mentions of potential negative societal impacts, but this is fairly abstract work. As an exercise, the authors may want to situate the impact of RL as a field (to which they contribute to) on society.

**Main Review:**

Although it proposes an interesting method, this work does not really prove its hypothesis.
The central hypotheses of this work are that (A) RL methods are susceptible to target noise and (B) LF2 reduces noise in the Bellman update. (A) is used as a premise here and is never shown to be true, and although it is "common wisdom" in the deep RL community, it is contrary to some existing empirical literature. (B) is also never measured directly, and so never shown to be true.
While it appears correct to claim that LF2 improves performance in some cases, it is impossible to link that improvement to the noise reduction hypothesis.

The theoretical part of this paper is interesting, and is probably a valuable contribution. Unfortunately, results in this section do not trivially transfer to hypotheses (A) & (B).

While I am confortable with deep RL, I am only superficially familiar with Fourier features literature. Training Fourier features appears to be something that isn't normally done, and while lots of papers have proposed regularizations to value functions, I am unaware of frequency-domain methods. The paper itself is well written and was easy to understand.

Although I find the method interesting and in-line with many intuitions that deep RL researchers have, this paper does not teach us much, and relies too heavily on proxies to "prove" its hypothesis. More work needs to be done for this paper to be impactful, and to really teach us something about deep RL.


Specific comments:
- l.32 Farebrother et al. [1] successfully apply dropout to DQN
- Section 3, Some unusual notation: using \gamma for the frequency mapping, and \tau for the activation function. In RL, \phi is typically used for the former, and in deep learning, \sigma or `a` for the latter.
- l.245 'with the shaded region denoting 68% confidence interval'. 68%? This seems like a rather odd choice. 90 and 95% confidence intervals are more usual. I wonder what justified this choice? You may want to refer to papers such as [2] for guidance.
- l.248 'LF2 does clearly better than MLPs in 6 out of 8 environments'. My _subjective_ reading of Figure 5 does not suggest so, but a statistical test would provide a quantitative assessment of superiority. How statistically significant are these improvements?
- Figure 8, I again doubt that any of these curves are significantly different from each other.
- l.285 'This validates our hypothesis that Bellman updates remain noisy [..] and that LF2 reduces [noise]'. I'm not convinced by this experiment. Not only are the results of Figure 8 probably marginal, noise has not been measured. It would be much more convincing to actually measure this noise, if it exists at all (current literature suggests instead that deep value networks _underfit_ severely, thus what is badly propagated is not noisy memorized estimates but underfit overgeneralized value estimates, see [3]; [6] may be relevant as well, and touches on the correlation between Bellman noise and agent performance in Atari).
- I find the choice of using SAC as a baseline to improve over quite odd. Presumably, the proposed method stabilizes the value function, but SAC is a policy gradient method -- one can only expect marginal improvements from improving the value function. Why not compare to a value-based method?
- Baselines: A number of functional regularizations for deep RL have been proposed recently, but none appear here. For references, see e.g. [3,4] and Section 5 of [5].


[1] Generalization and Regularization in DQN, Jesse Farebrother, Marlos C. Machado, Michael Bowling, 2018
[2] Deep Reinforcement Learning that Matters, Henderson et al. 2017
[3] Implicit Under-Parameterization Inhibits Data-Efficient Deep Reinforcement Learning, Aviral Kumar, Rishabh Agarwal, Dibya Ghosh, Sergey Levine, 2020
[4] Temporal Regularization in Markov Decision Process, Pierre Thodoroff, Audrey Durand, Joelle Pineau, Doina Precup, 2018
[5] Beyond Target Networks: Improving Deep Q-learning with Functional Regularization, Alexandre Piché, Joseph Marino, Gian Maria Marconi, Christopher Pal, Mohammad Emtiyaz Khan, 2021 (Yes, this paper was only recently submitted to arxiv, but its Section 5 might contain good works to refer to)
[6] Interference and Generalization in Temporal Difference Learning, Bengio et al. 2020


=======
Update
I thank the authors for their answers, updates and additional experiments, I think that these more convincingly explain why the proposed method is advantageous. My main problem with the paper was connecting the hypothesis behind the method with the improvements, but this connection is much better now. I will update my score. To be clear, validating all hypotheses should be a main feature of papers like this one, not just showing improvements. I encourage the authors to feature this prominently when updating this paper.
It does additionally seem like the improvements in the paper are consistent. Their magnitude is limited, but could be valuable in certain environments where target noise can become a significant issue.

**Time Spent Reviewing:**

4

---

> ### Author Response · Authors · 2021-08-11
> **Response to Reviewer # Susc**
>
> Dear Reviewer,
>
> Thank you for the helpful and insightful feedback! We believe we have addressed all the reviewer’s concerns below. If there are any further concerns that prevent you from accepting the paper, please let us know and we will attempt to address them!
>
> > *l.32 Farebrother et al. [1] successfully apply dropout to DQN*
> - Upon reviewer’s suggestion, we ran a comparison to dropout, where we add a dropout layer after every nonlinearity in the MLP. We search over [0.05, 0.2] for the drop probability, and find that lower is better. Link to dropout results: https://drive.google.com/file/d/1iaIiTHX6T93ZV17SMYRrYaL6VgkIaBX5/view?usp=sharing
> - Our results show that dropout does help in most of the DMControl environments, although occasionally at the cost of asymptotic performance. Still, LF2 still consistently ranks around the top across all of the environments, and can be combined with dropout for more potential efficiency gains.
> - A really interesting observation is that dropout can be thought of as explicit smoothing. At each bootstrap iteration, a different random Q-net subnetwork is fit to noisy/inaccurate target values. When using the Q-function (i.e. when generating target values or optimizing the policy), we use eval mode and use the full network, which is roughly akin to averaging over all possible subnetworks and their outputs. Thus, dropout is explicitly smoothing the noise fit by each random subnetwork, whereas LF2 implicitly smooths the noise by fitting it extremely slowly. We’ll add this cool connection to the paper, and also make clear that dropout is actually quite useful in deep RL.
>
> > *I find the choice of using SAC as a baseline to improve over quite odd. Presumably, the proposed method stabilizes the value function, but SAC is a policy gradient method -- one can only expect marginal improvements from improving the value function.*
> - It is true that LF2 shouldn’t help much with policy gradient methods like PPO and we do mention this in Appendix Section E and Appendix Fig. 5. However, SAC is not a policy gradient method. It is an off-policy actor-critic method that heavily relies on accurate Q-value estimates. LF2 helps here because it stabilizes the bootstrapping process of training the Q-functions to minimize the Bellman error.
>
> > *Baselines: A number of functional regularizations for deep RL have been proposed recently, but none appear here. For references, see e.g. [3,4] and Section 5 of [5].*
> - Thank you -- we will add these papers to the Related Works section.
> - Additionally, based on the reviewer’s suggestion, we ran functional regularization [5] on the state-space DM Control environments, and found that functional regularization often helps, but LF2 is still better on 4/7 of the environments tried, and the same on 2/7. Link to functional regularization results: [https://drive.google.com/file/d/1iaIiTHX6T93ZV17SMYRrYaL6VgkIaBX5/view?usp=sharing](https://drive.google.com/file/d/1iaIiTHX6T93ZV17SMYRrYaL6VgkIaBX5/view?usp=sharing).
> - Notably, LF2 and FR have different behavior. This is because LF2 is biased towards fitting low-frequency TD error residuals, without regard to the norm, whereas FR fits residuals of any frequency, as long as their norm is not too large. Combining these two methods may be useful for handling both cases at once.
>
> > *l.285 'This validates our hypothesis that Bellman updates remain noisy [..] and that LF2 reduces [noise]'. I'm not convinced by this experiment. Not only are the results of Figure 8 probably marginal, noise has not been measured. It would be much more convincing to actually measure this noise, if it exists at all (current literature suggests instead that deep value networks underfit severely, thus what is badly propagated is not noisy memorized estimates but underfit overgeneralized value estimates, see [3]; [6] may be relevant as well, and touches on the correlation between Bellman noise and agent performance in Atari).*
> - It’s difficult to measure the noise in the target values, as (a) it’s unclear what the ground truth target values should be in the middle of training, and (b) the Q-functions in deep RL do not tend to converge to the same values by the end of training.
> - However, to address the reviewer's concern, we ran an experiment where we add zero-mean Gaussian noise to the targets and compare LF2 to MLPs. We keep target networks for LF2 and MLP in this experiment. We see that LF2 generally does much better than MLPs, and can maintain decent performance even under significant amounts of noise (e.g. standard deviation of 30). Link to noise experiment results: [https://drive.google.com/file/d/1rwLTr-AyY10yo4D3mBULQuey5JLqHLnm/view?usp=sharing](https://drive.google.com/file/d/1rwLTr-AyY10yo4D3mBULQuey5JLqHLnm/view?usp=sharing).
> - We believe that [3] does not convincingly show that underfitting is happening in off-policy RL. It uses `srank` (which roughly measures how equal the singular values of the last layer are) as a proxy for underfitting, but it’s unclear how good this measure is. One can find poor representations that cause underfitting but have high `srank`, or good representations (e.g. the representation is just the Q-value, followed by D-1 zeros) with low `srank`. Most of the experiments have the number of gradient steps as a major confounding factor. Training longer decreases `srank`, but it’s unclear if the performance drop is caused by low `srank`, or if it’s because more gradient steps propagate and amplify noise, and the `srank` decreases as a side effect. Figures A15-A17 of [3] show that, even as `srank` increases with the proposed penalty, the performance drops with more gradient steps, which indicates that the number of updates is a huge confounding factor. The proposed penalty is actually consistent with overfitting to noise -- penalizing the difference between the largest and smallest singular values is akin to spectral normalization, which helps encourage smoothness. This matches our hypothesis that  noise in the bootstrap process is present and harmful to performance.
>
> > *Section 3, Some unusual notation: using \gamma for the frequency mapping, and \tau for the activation function. In RL, \phi is typically used for the former, and in deep learning, \sigma or a for the latter.*
> - We chose our notation to maintain consistency with prior work that we draw on. Tancik et al use \gamma for the frequency mapping and \sigma for the initialization variance. Thus, we picked \tau for the activation function. We hope that, while unusual for a RL paper, our notation is clear enough to readers.
>
> > *l.245 'with the shaded region denoting 68% confidence interval'. 68%? […] 90 and 95% confidence intervals are more usual. [...] You may want to refer to papers such as [2] for guidance.*
> - Sorry for the terminology confusion here. The shaded confidence interval in the plots of our paper is exactly the “standard error of mean” which is a common way to show the error bars in machine learning  (e.g. in [2]). We will reword the phrase in the paper to avoid this confusion for readers.
>
> > *l.248 'LF2 does clearly better than MLPs in 6 out of 8 environments'. My subjective reading of Figure 5 does not suggest so, but a statistical test would provide a quantitative assessment of superiority.*
> - We had been looking only at LF2 vs vanilla MLPs, but will change the claim to reflect LF2 performance vs the best baseline in each separate environment. LF2 is still around the top for every environment, and is compatible with weight decay and the other regularization methods we added in reply to Reviewer 3 # aHKm, so performance may further increase when LF2 is combined with these methods.
>
> > *Figure 8, I again doubt that any of these curves are significantly different from each other.*
> - We agree that some of these curves do overlap significantly, e.g. every curve in Finger Turn, or LF2 policy (red) and MLP (orange) in Quadruped Run. Nonetheless, the standard error is quite low, so we believe the general conclusion (regularizing the Q-function with LF2 is what improves performance) still holds. Please also refer to the several new experiments we have added in reply to Reviewer 3 # aHKm.

---

> > ### Comment · Reviewer_Susc · 2021-08-13
> > **Updates**
> >
> > Thank you for your answers, updates and additional experiments, I think that these more convincingly explain why the proposed method is advantageous. My main problem with the paper was connecting the hypothesis behind the method with the improvements, but this connection is much better now. I will update my score.
> > To be clear, validating all hypotheses should be a main feature of papers like this one, not just showing improvements. I encourage the authors to feature this prominently when updating this paper.
> >
> > > It’s difficult to measure the noise in the target values
> >
> > I agree, although I think infirming or confirming that this is a real thing in RL and quantifying it would be an important contribution. So far all of the literature only provides circumstantial evidence.
> >
> > > it’s unclear what the ground truth target values should be in the middle of training, the Q-functions in deep RL do not tend to converge to the same values by the end of training
> >
> > It's likely that we wouldn't need ground truth in order to establish this. What's really necessary is to establish the noise of the (non-linear) correlation between input and target, _during_ training rather than at convergence. Food for thought.
> >
> > > However, SAC is not a policy gradient method
> >
> > I guess you're technically correct, but SAC remains an instance of soft policy iteration, whereby the main parameterization being driven is a policy (admittedly driven by Q). I think testing this on a "pure" value-based method could be informative, it wouldn't even need to be a control experiment, policy evaluation would be convincing here.

---

> > > ### Author Response · Authors · 2021-08-16
> > > **Response and New Experiment**
> > >
> > > Thank you for following up on the discussion! We will be sure to feature the new experiments prominently. Sorry for the small delay in reply, as we were running the experiment to quantify noise in off-policy RL on a controlled environment per your suggestion. We are pleased to report that we have finished it and obtained results consistent with our hypothesis (see below).
> > >
> > > > *“I think infirming or confirming [noise in the target values] is a real thing in RL and quantifying it would be an important contribution. So far all of the literature only provides circumstantial evidence. [...] I think testing this on a "pure" value-based method could be informative”*
> > >
> > > - Thank you for the suggestion! To precisely test whether LF2 helps by stabilizing the learned Q-function, we ran an additional experiment to demonstrate noise amplification in a controlled setting. We use the 16x16 gridworld from Kumar et al [3], which allows us to use Q-iteration to find the optimal Q* values. We train vanilla MLPs or LF2 using fitted Q-iteration over a replay buffer, and compare the learned Q-functions to Q* over the course of training. This is a pure value-based method, as we only learn a Q-function; to get the policy, we took the argmax over the 5 available actions.
> > > - We see that the MLP’s Q-values grow exponentially in the first 100 iterations, far past what the correct values should be. The MLP’s target values grow in lockstep, indicating that a positive feedback loop drives both of them upwards. This effect is exacerbated by more gradient steps, and drives the MLP MSE with respect to Q* into the 10^7 range. In contrast, LF2 is very stable and shows little to no noise amplification. This further supports our hypothesis that LF2 improves off-policy RL by stabilizing the bootstrapping process. We will include this experiment in the paper. Link to the plots for the new noise amplification experiments: https://drive.google.com/file/d/12U-7ZVbTP8-bVhOcgk2Lb0Iumn0Zn6i_/view?usp=sharing

---

### Official Review · Reviewer_aHKm · 2021-07-12

**Rating:** 6
**Confidence:** 4

**Summary:**

The paper proposes to use input features concatenated with learned Fourier feature (LF2) basis to control the degree of fitting various frequencies in the training data. Theoretical analysis under NTK regime with 2 layer network shows that the initial variance of Fourier basis controls this degree. In value-based RL, the updates using target Q-values can result in noisy optimization and LF2 can reduce dependence on noise by prioritizing low-frequency functions.  Empirically, LF2 leads to small sample-efficiency gains on a subset of DM control environments.

**Limitations And Societal Impact:**

-- Small gains in empirical experiments
-- Lack of evaluation of relevant baselines listed in main review (such as deeper nets / spectral normalization / l2-normalization)
-- Adhoc selection of environments for reporting gains

====================================================================================================
Update

Based on the author response, I have updated the score to 6 (from initial score of 5) and increased confidence to 4 (from 3) as the new experiments and ablations show the proposed method indeed leads to consistent performance gains even though they may not be large. The authors also provided some empirical experiment on a toy gridworld domain, thus providing some evidence the premise of the paper for noise amplification in RL. Assuming that authors would incorporate the discussion in their camera ready (especially the part related to underfitting and noise amplification), I can recommend publishing this work at NeurIPS.

**Main Review:**

**Originality**: The paper can be considered as a small modification on top of existing ideas. Building on prior work[1] that introduced a fixed fourier feature basis to capture high frequency functions, the paper proposes LF2 which learns the fourier feature basis  as randomly initialized features can be quite limited. Additionally, it utilizes the architecture trick to concatenate the input features with these fourier features.

Although the related work is cited generally well, the paper doesn’t have an explicit section but could benefit from one. One point which is not discussed much is why the inherent spectral bias of deep networks towards learning low-frequency functions [1] is not sufficient for learning in deep RL. For example, recent work shows that this spectral bias can lead to too low effective rank of the learned features in deep RL [2] with theoretical results in the NTK regime. [3] also confirmed such findings for online RL.
While Figure 2 takes an simulation attempt at it but can possibly be alleviated with deeper networks like ResNets or just using L2-regularization. Also

**Quality**: The submission seems to be technically sound. I did not go through the proofs but the lemme presented in the paper made sense. The empirical evaluation currently seems to indicate LF2 only results in tiny improvements and could be somewhat stronger. Some concerns I have are listed below:

- The empirical results in Figure 5 indicate that LF2 gains are marginal compared to standard MLPs (and might have just resulted from randomness in evaluation on some tasks). Furthermore, l2-weight decay performs reasonably well on most of the domains (and even outperforms LF2 on certain domains). Same is found to be true for PPO-based agents in the appendix.

- The results on image-based domains in Figure 6 seem moderate too and are evaluated only in a smaller subset of 4 environments reported in Figure 5 (which may seem to be cherry-picked?).
Furthermore, no other baseline is evaluated on such image-based domains including the l2-reg baseline in Fig 5. If the improvement is coming from learning smoother functions, another way to impose such constraints is to impose spectral normalization which is also reported to work quite well in RL [3] (and even on DM control tasks[4]).

- The ablation study in Sec 6.2 cripples the standard SAC agent by removing the target network and adding a noise value of “11” and shows that LF2 works well. However, in practice, target networks are almost always used. So, how does LF2 compare to standard SAC when the target network is added back – are target networks sufficient for handling the noise? Also, why was a specific noise value of 11 picked – why not smaller/larger (or better a range of values on log scale)?

- How much of the improvement in LF2 is just coming from concatenation of the input?  It was not clear to me whether the ablation studies answer this question (as there was no baseline which uses concatenated inputs with MLP/conv architecture with the same parameter count).
Specifically, what if I use the same architecture for the baseline as used currently but with the inputs concatenated on the first layer output as done for the LF2 architecture?

**Clarity**: The submission seems to be well written and organized. A minor comment: Instead of reporting the 68% confidence intervals, it might be better to report the standard deviation as 68% CIs may not have a true coverage of 68% as the reported results are only with 5 seeds.

**Significance**: The results seem to indicate that LF2 architecture may be better than standard MLPs on certain hard tasks such as humanoid. However, it is unclear when LF2 should be used or why it should be preferred over other existing methods like using ResNets or spectral normalization. That said, the example pointed in the appendix about learning overly complex Q-functions was quite neat and shows that LF2 might be useful in certain scenarios (but the main paper focuses on the other scenario). Overall, the paper in its current form may not be that exciting to the community.

[1] Rahaman, Nasim, et al. "On the spectral bias of neural networks." International Conference on Machine Learning. PMLR, 2019.
[2]. Kumar, Aviral, et al. "Implicit Under-Parameterization Inhibits Data-Efficient Deep Reinforcement Learning." International Conference on Learning Representations. 2020.
[3] Gogianu, Florin, et al. "Spectral Normalization for Deep Reinforcement Learning: an Optimization Perspective." ICML (2021).
[4] Bjorck, Johan, Carla P. Gomes, and Kilian Q. Weinberger. "Towards Deeper Deep Reinforcement Learning." arXiv preprint arXiv:2106.01151 (2021).

**Time Spent Reviewing:**

4.5

---

> ### Author Response · Authors · 2021-08-11
> **Response to Reviewer # aHKm**
>
> Dear Reviewer,
>
> Thank you for the helpful and insightful feedback! We believe we have addressed all your questions below and ran all the experiments you suggested. If there are any further concerns that prevent you from accepting the paper, please let us know and we will attempt to address them in the follow up!
>
> >*One point which is not discussed much is why the inherent spectral bias of deep networks towards learning low-frequency functions [1] is not sufficient for learning in deep RL.*
> - This is a good question! We discussed this briefly in Section 3. On line 80, we write: “The problem is that MLPs provide no control over how quickly they learn signals of different frequencies.” To further clarify, the inherent spectral bias of MLPs likely helps stabilize deep RL bootstrap updates to some degree -- we want to greatly increase this spectral bias towards low frequencies by using learned Fourier features. Figure 4 of the original submission compares the eigenvalues (and thus the exponential rate of per-frequency convergence) of MLPs to LF2 with various initialization variances. MLPs are biased towards learning low-frequency functions faster, but we can accentuate this effect by using small sigma, which helps to further stabilize the Bellman updates. We will add this explanation in the paper.
>
> > *For example, recent work shows that this spectral bias can lead to too low effective rank of the learned features in deep RL [2] with theoretical results in the NTK regime. [3] also confirmed such findings for online RL.*
> - Spectral bias is related to, but isn’t the same thing as low effective rank in RL. Both are analyzed with the NTK, but spectral bias is present in supervised learning and RL, whereas low rank only affects self-training and bootstrapping.
> - We believe that [2] does not convincingly show that underfitting is happening in off-policy RL. It uses `srank` (which roughly measures how equal the singular values of the last layer are) as a proxy for underfitting, but it’s unclear how good this measure is. One can find poor representations that cause underfitting but have high `srank`, or good representations (e.g. the representation is just the Q-value, followed by D-1 zeros) with low `srank`. Most of the experiments have the number of gradient steps as a major confounding factor. Training longer decreases `srank`, but it’s unclear if the performance drop is caused by low `srank`, or if it’s because more gradient steps propagate and amplify noise, and the `srank` decreases as a side effect. Figures A15-A17 of [3] show that, even as `srank` increases with the proposed penalty, the performance drops with more gradient steps, which indicates that the number of updates is a huge confounding factor.  The proposed penalty is actually consistent with overfitting to noise -- penalizing the difference between the largest and smallest singular values is akin to spectral normalization, which helps encourage smoothness. This matches our hypothesis that  noise in the bootstrap process is present and harmful to performance.
>
> > *If the improvement is coming from learning smoother functions, another way to impose such constraints is to impose spectral normalization which is also reported to work quite well in RL [3] (and even on DM control tasks[4]).*
> - Based on the reviewer’s suggestion, we tried adding spectral normalization to the second-to-last layer of the network, as is done in [3]. We find that this works very poorly on state-space experiments with SAC. It’s likely necessary to tune the other hyperparameters (learning rate, target update frequency, Polyak averaging parameter) in order to make spectral normalization work. This is why LF2 has a plug-and-play advantage -- one setting for the fourier_dimension and sigma works well across many environments with the default SAC hyperparameters tuned for MLPs. Link to the plots with these added baselines: [https://drive.google.com/file/d/1iaIiTHX6T93ZV17SMYRrYaL6VgkIaBX5/view?usp=sharing](https://drive.google.com/file/d/1iaIiTHX6T93ZV17SMYRrYaL6VgkIaBX5/view?usp=sharing).
>
> > *The ablation study in Sec 6.2 cripples the standard SAC agent by removing the target network and adding a noise value of “11” and shows that LF2 works well. However, in practice, target networks are almost always used. So, how does LF2 compare to standard SAC when the target network is added back – are target networks sufficient for handling the noise? Also, why was a specific noise value of 11 picked – why not smaller/larger (or better a range of values on log scale)?*
> - We apologize for casual usage of the phrase: “crank the noise up to 11” -- this is a colloquial American phrase which basically means amplifying the noise. We will remove this line from the paper. To clarify, for Section 6.2/Fig. 7, we simply removed the target networks, and examined how LF2 and MLPs do in their absence.
> - Furthermore, to specifically test the hypothesis that LF2 does better because of improved stability properties, we ran a new experiment where we added zero-mean Gaussian noise to the target values and compared LF2 vs MLP in this setting. The results are here: [https://drive.google.com/file/d/1rwLTr-AyY10yo4D3mBULQuey5JLqHLnm/view?usp=sharing](https://drive.google.com/file/d/1rwLTr-AyY10yo4D3mBULQuey5JLqHLnm/view?usp=sharing). For this experiment, we keep the target networks for both LF2 and MLPs. We see that LF2 generally does much better than MLPs, and can maintain decent performance even under significant amounts of added noise (e.g. standard deviation of 30). This experiment precisely shows that LF2 tends to be more stable than MLPs, and we can extrapolate from this to infer that stability is at least partially why LF2 does better than MLPs when no noise is artificially added.
>
> > *How much of the improvement in LF2 is just coming from concatenation of the input? [...] what if I use the same architecture for the baseline as used currently but with the inputs concatenated on the first layer output as done for the LF2 architecture?*
> - Thanks for suggesting this ablation! We ran this and found that there was no significant benefit when concatenating the input to the MLP’s first layer output. Link to this new result is here: https://drive.google.com/file/d/1j98y78nP5U6tKJt1TtQ2R4vgzI5cOuEr/view?usp=sharing
>
> >*Furthermore, no other baseline is evaluated on such image-based domains including the l2-reg baseline in Fig 5.*
> - As suggested by the reviewer, we added the L2 baseline to the 4 image-based environments, and found that this greatly hurt sample efficiency and asymptotic performance, even for very low weight decay settings (1e-5). Link to L2 baseline with image-based results: https://drive.google.com/file/d/1coSuLkYR9dr1JbLYrwNCcz3IeWddYrZS/view?usp=sharing
>
> > *The empirical results in Figure 5 indicate that LF2 gains are marginal compared to standard MLPs (and might have just resulted from randomness in evaluation on some tasks). Furthermore, l2-weight decay performs reasonably well on most of the domains (and even outperforms LF2 on certain domains). Same is found to be true for PPO-based agents in the appendix.*
> - Compared to regularization methods like WD, or new baselines we’ve added on suggestion from other reviewers (spectral normalization, dropout, and functional regularization), LF2 consistently ranks around the top in every environment. Link here: [https://drive.google.com/file/d/1iaIiTHX6T93ZV17SMYRrYaL6VgkIaBX5/view?usp=sharing](https://drive.google.com/file/d/1iaIiTHX6T93ZV17SMYRrYaL6VgkIaBX5/view?usp=sharing). More importantly, LF2 is complementary to and can be combined with any of these regularization methods, if desired. Finally, LF2 is very easy to use -- one set of hyperparameters work consistently across all state-based environments. Thus, we believe that LF2 could be a good plug-and-play replacement for vanilla MLPs for off-policy deep RL.
> - PPO experiment: LF2 helps with off-policy RL by stabilizing the Bellman update. Hence, the point of the PPO experiment was precisely to show that the LF2 shouldn’t help much because PPO is less reliant on accurate value functions, since it (a) uses multi-step returns and (b) only uses the value functions as a baseline to reduce variance.  We will make it more clear that the lack of significant improvement for PPO is the expected outcome.
>
> > *The results on image-based domains in Figure 6 seem moderate too and are evaluated only in a smaller subset of 4 environments reported in Figure 5 (which may seem to be cherry-picked?).*
> - State-based: Due to our limited compute budget, we decided to not use DMControl environments that are too easy and are basically fully solved by any method. Instead, we picked the environments which are commonly evaluated in RL papers and are present in both DMControl as well as OpenAI Gym -- i.e. Cheetah, Walker, Hopper, Acrobot, Quadruped, and Humanoid.
> Image-based: SAC is much slower to train for image-based methods. Hence, out of the above 8, we picked the four *moderately difficult* environments, discarding the ones which were too easy from state-space. We also left out Humanoid because no image-based method we tried could make progress on it (note: most DMControl image-based RL papers don’t show Humanoid). We are currently evaluating more image-based environments and will post results once the experiments finish.

---

> > ### Comment · Reviewer_aHKm · 2021-08-12
> > **Reviewer response to author updates**
> >
> > Thanks for running the additional baselines, ablation and providing clarifications -- I believe they would help improve the significance and possible impact of the paper. I am still unsure about their response to underfitting in deep RL vs noise amplification (also raised by reviewer Susc):
> >
> > >  Training longer decreases srank, but it’s unclear if the performance drop is caused by low srank, or if it’s because more gradient steps propagate and amplify noise, and the srank decreases as a side effect. Figures A15-A17 of [3] show that, even as `srank` increases with the proposed penalty, the performance drops with more gradient steps, which indicates that the number of updates is a huge confounding factor.
> >
> > Indeed, `srank`  is not a cause of performance degradation in offline/data-effiicient RL settings and only claimed to be correlated with such degradations in [2]. Note that significant `srank` drops require a large number of gradient steps akin to running self-distillation for a large number of updates. That said, I am not sure whether I agree with the authors' response that low srank doesn't imply underfitting -- low `srank` can be interpreted as low expressivity of the features -- there are experiments in [2] which regressed the optimal Q-functions using supervised learning with such features and show that a lower `srank`  correspond to a poorer fit.  Now, the response mentions that noise amplification is different than underfitting -- it is unclear to me if that's the case and whether we can actually measure this "amplification". Can the authors expand on this point (or maybe provide empirical justification)?
> >
> > > Figures A15-A17 of [3] show that, even as srank increases with the proposed penalty, the performance drops with more gradient steps, which indicates that the number of updates is a huge confounding factor. The proposed penalty is actually consistent with overfitting to noise -- penalizing the difference between the largest and smallest singular values is akin to spectral normalization, which helps encourage smoothness. This matches our hypothesis that noise in the bootstrap process is present and harmful to performance
> >
> > I am not sure which figures are being referenced by the authors -- the discussion of `srank` in [3] is done in appendix C.4 where they show that `srank` decreases as training progresses while the proposed spectral normalization in [3] stabilizes the srank.
> >
> > If the authors are referring to figures in [2], it seems that authors presented their penalty to increase `srank`. Can the authors clarify how a high `srank` (let's say all the singular values of the feature matrix are equal) corresponds to smoothness?

---

> > > ### Author Response · Authors · 2021-08-16
> > > **Response and New Experiment**
> > >
> > > Dear Reviewer,
> > >
> > > Thank you for your thoughtful discussion. Sorry for our delayed response, as we were running the experiment to measure noise “amplification” per your suggestion. We are pleased to report that we have finished it and obtained results consistent with our hypothesis (see below). Please let us know if you have more thoughts/questions. If you believe your concerns are answered, we would love to know that as well.
> > >
> > > > *unclear to me if that's the case and whether we can actually measure this "amplification". Can the authors expand on this point (or maybe provide empirical justification)?*
> > > - We think much of this debate is over the term “underfitting” whose definition is not very clear when it refers to matching moving target values. To us, a Q-network underfits when it does not have enough capacity to match *fixed* target values within a reasonable number of gradient steps. This is different from noise amplification, which is when updates to the Q-network at (s, a) unintentionally change the values at a different (s’, a’). This changes the targets, which further changes Q(s, a), and the cycle continues. Thus, noise amplification and underfitting could both simultaneously occur.
> > > - As per reviewer’s suggestion, we run an additional experiment to demonstrate noise amplification in a controlled setting. We use the 16x16 gridworld from Kumar et al [2], which allows us to use Q-iteration to find the optimal Q* values. We train vanilla MLPs as well as LF2 using fitted Q-iteration over a replay buffer, and compare the learned Q-functions to Q* over the course of training. We see that the MLP’s Q-values grow exponentially in the first 100 iterations, far past what the correct values should be. The MLP’s target values grow in lockstep, indicating that a positive feedback loop drives both of them upwards. This effect is exacerbated by more gradient steps, and drives the MLP MSE with respect to Q* into the 10^7 range. In contrast, LF2 is very stable and shows little to no noise amplification. This further supports our hypothesis that LF2 improves off-policy RL by stabilizing the bootstrapping process. We will include this experiment in the paper. Link to the plots for the new noise amplification experiments: https://drive.google.com/file/d/12U-7ZVbTP8-bVhOcgk2Lb0Iumn0Zn6i_/view?usp=sharing
> > >
> > > >*I am not sure which figures are being referenced by the authors*
> > > - Yes, sorry, you are right. We were referring to the figures in Kumar et al. ICLR’21 [2], not [3]. It was a typo.
> > >
> > > > *If the authors are referring to figures in [2], it seems that authors presented their penalty to increase srank. Can the authors clarify how a high srank (let's say all the singular values of the feature matrix are equal) corresponds to smoothness?*
> > > - To clarify, we did not mean that high srank implies smoothness. For example, consider the case when all singular values are equal to some large number. In this case, the srank is high, but the feature matrix will also have high magnitude and vary wildly between inputs. Hence, the changes in input will lead to large changes in output meaning the network has a very large Lipschitz constant and is not smooth. On the other hand, if the magnitude of maximum singular value is low, that could imply a small Lipschitz constant (holding the weight matrix constant), and therefore, more smoothness.
> > > - To further clarify, below is the reason we believe that the proposed penalty in Kumar et al. ICLR’21 [2] might be achieving high srank by actually encouraging smoothness. Consider Figure D.1 in Kumar et al. ICLR’21 [2] which shows that the largest singular value is multiple orders of magnitude larger than the smallest ones. In their penalty $\sigma_{max}^2 - \sigma_{min}^2$, the first term is extremely large, while the second term is small. Thus, there is immense downward pressure on the largest singular value, and small upwards pressure on the smallest singular value. As a result, the maximum singular value should in theory become far smaller (Kumar et al did not show the effect of their penalty on the singular values), which could lead to the representations that are smoother, as we discussed above.

---

> > > > ### Comment · Reviewer_aHKm · 2021-08-16
> > > > **Thanks for the clarifications!**
> > > >
> > > > Thanks for the additional experiment (one thing that would further provide evidence for your hypothesis is to also report `srank` in the new experiment added above to indicate whether underfitting in form of low `srank` and noise amplification are happening simultaneously or not, my guess is that they are -- I would be curious to see those results too). Overall, I feel that this paper can now be published at NeurIPS.

---

> > > > > ### Author Response · Authors · 2021-08-17
> > > > > **Thank you! (+ new experiment with srank)**
> > > > >
> > > > > > **Overall, I feel that this paper can now be published at NeurIPS.**
> > > > > - We are very happy to hear this and have enjoyed our discussion with the reviewer. We kindly request the reviewer if they could update their main review/score to reflect this final opinion so that the area chair can easily see this.
> > > > >
> > > > > > *one thing that would further provide evidence for your hypothesis is to also report srank in the new experiment added above to indicate whether underfitting in form of low srank and noise amplification are happening simultaneously or not*
> > > > > - We reran the Gridworld Fitted Q-iteration experiment above and logged the `srank`, maximum singular value of the feature matrix, and minimum singular value of the Q-network’s feature matrix. Surprisingly, we find that noise amplification in MLPs happens even as `srank` remains high. This implies that underfitting (at least the kind correlated with `srank`) is not happening simultaneously for MLPs. Overall, this experiment provides evidence that noise amplification can happen without underfitting. We will include this experiment in the paper. Link to the plots for the new `srank` experiment: https://drive.google.com/file/d/1qLk0OUUbkL-nmO4m-hJAvEva5I5I4oBQ/view?usp=sharing
> > > > >
> > > > > Thank you again for the insightful feedback and discussion.

---

### Official Review · Reviewer_jV7a · 2021-07-16

**Rating:** 6
**Confidence:** 3

**Summary:**

In this work, the authors analyze a layer for deep networks based on learnable Fourier features and propose to employ it as a way to improve performance in architectures used for Deep Q-Learning (DQL). Specifically, they introduce this layer to control the noise as a function of the frequency in the input domain, and parametrize the granularity of such features by means of a learnable parameter.

The authors analyze the behaviour of such layer using a neural tangent kernel approximation and empirically evaluate its noise-filtering properties in various DQL experiments, showing its benefits to the task.
One of the main claims of the paper is that by controlling the variance of the stochastic initialization of the proposed layer, they can control the frequencies at which the noise will be filtered, with subsequent improvements in terms of sample complexity and stability.

**Limitations And Societal Impact:**

I think the authors could stress more the limitation of their analysis and that it's just a sketch mostly caused by the current state of theory in DL.

**Main Review:**

While the idea of using base layers that learn a representation in some harmonic domain is not new (e.g. Alexandridis 2013, Zuo 2008,  Oyallon 2014-2017), to the best of my knowledge no prior application to target networks is present in the literature. Still, I suggest to expand the introduction with some of this references, mentioning that the base idea is not new.

While the theoretical analysis is interesting in principle, it does not offer a strong insight on the proposed method. Specifically, Neural Tangent Kernel analysis makes the assumption that the network does not change significantly from the initialization, which is in-line with the idea that the initialization variance of the Fourier layers matters. However, if that is true, is it even worth training those layers? I invite the authors to investigate more the impact of early layers focusing on certain representations, either fixed or learned.
Another suggestion would be to use alternative tools for analyzing the behaviour of the network, not at initialization but at convergence (e.g. Approximate Inference Turns Deep Networks into Gaussian Processes - 2018), or studying the stability of the Bellman operator when using such layers (e.g. see Towards Characterizing Divergence in Deep Q-Learning - 2019).

While the analysis has shortcomings, the proposed approach is investigated thoroughly in the experimental section, which constitutes the more relevant part of the work. The experimental set-up look adequate and the results are good, albeit not groundbreaking in terms of performance. I appreciate that the authors not only validated the overall performance in terms of reward but also the stability during training.

Overall, the paper is clear and well-structured. Because I deem the fundamental questions investigated by this paper to be very relevant for the RL community, either in theoretical or empirical form, I think its shortcomings could be overshadowed by its positive effect on the RL community.


**Time Spent Reviewing:**

3

---

> ### Author Response · Authors · 2021-08-11
> **Response to Reviewer # jV7a**
>
> Dear Reviewer,
>
> Thank you for the helpful and insightful feedback! We respond to the reviewer’s comments below.
>
> > *Neural Tangent Kernel analysis makes the assumption that the network does not change significantly from the initialization, which is in-line with the idea that the initialization variance of the Fourier layers matters. However, if that is true, is it even worth training those layers? [...] I think the authors could stress more the limitation of their analysis and that it's just a sketch mostly caused by the current state of theory in DL.*
> - Yes, it is important to train the layers -- the NTK analysis only assumes that the change in norm of the layer weights is very small compared to the original norm of the layer weights. We do still need to train the layers in order to learn. However, we do agree with you that NTK makes several unrealistic assumptions, especially for convolutional neural networks [1]. We will make clear the limitations of NTK analysis in the main text.
> - That being said, NTK analysis matches up quite well with empirical behavior in our case. For example, it predicts the frequency-dependent behavior of LF2 with varying sigma in Figure 1, 2, and Appendix Fig. 6, especially compared to vanilla MLPs. This is likely because we use pretty wide networks (hidden dimension of 1024), so the kernel function is close to the infinite-width limit (Figure 3b).
>
> > *Another suggestion would be to use alternative tools for analyzing the behaviour of the network, not at initialization but at convergence (e.g. Approximate Inference Turns Deep Networks into Gaussian Processes - 2018), or studying the stability of the Bellman operator when using such layers (e.g. see Towards Characterizing Divergence in Deep Q-Learning - 2019).*
> - Thank you for your suggestions! The first paper (Approximate Inference Turns Deep Networks into Gaussian Processes, 2018) analyzes network behavior at convergence, which is in contrast to our NTK analysis describing the training dynamics during learning and not just convergence. Our NTK analysis helps us understand that LF2 shines not because of its behavior at convergence, but because it controls the convergence rate for high frequencies.
> - The second paper (Towards Characterizing Divergence in Deep Q-Learning, 2019) is also NTK-based, and it is quite restrictive -- its analysis only works for discrete MDPs where we have a buffer of all the transitions. It also only examines conditions for when Bellman updates over the replay buffer are always contractions, which isn’t necessary for TD learning to eventually converge.
> - Nonetheless, based on the reviewer’s recommendation, we apply our LF2 NTK analysis in the setting from this second paper, and find that each discrete MDP has a minimum sigma value required for the Bellman update to always be a contraction. This setting matches up more with Appendix F (high frequency learning), as LF2 with large sigma values fit high frequencies quickly, and have very similar behavior as tabular functions. This new analysis is available here: [https://drive.google.com/file/d/1EXYky8xoOtbdISiJ5_mLrGXg4IReih04/view?usp=sharing](https://drive.google.com/file/d/1EXYky8xoOtbdISiJ5_mLrGXg4IReih04/view?usp=sharing)
>
> > *While the idea of using base layers that learn a representation in some harmonic domain is not new (e.g. Alexandridis 2013, Zuo 2008, Oyallon 2014-2017), to the best of my knowledge no prior application to target networks is present in the literature. Still, I suggest to expand the introduction with some of this references, mentioning that the base idea is not new.*
> - Thank you -- we will add these references!
> - Also, we just want to clarify: we apply LF2 to every network, not just the target network. In state-space SAC, this comprises Q-functions, target Q-functions, and policy; in image-based SAC, this is the convolutional encoder that feeds into the policy and Q-value heads.
>
> ### References
> [1] Allen-Zhu, Z., & Li, Y. (2020). Towards understanding ensemble, knowledge distillation and self-distillation in deep learning. arXiv preprint arXiv:2012.09816.

---

### Official Review · Reviewer_N6ix · 2021-08-03

**Rating:** 7
**Confidence:** 4

**Summary:**

This paper introduces a novel data transformation for deep reinforcement learning which enables faster learning of low-frequency information and empirically improves the performance of deep RL agents on 8 state-based DM control tasks and 4 image-based DM control tasks. The paper shows theoretically the convergence rate of the proposed transformation for neural tangent kernels to desired frequencies.

**Limitations And Societal Impact:**

The authors have sufficiently addressed both limitations and societal impact.

**Main Review:**

I recommend to accept this paper. [Str1] The proposed transformation has reasonable empirical support and is sufficiently general to likely be applicable to a wide variety of problems. [Str2] The goal of adjusting the learned representation of deep RL algorithms to avoid the need for additional stability tricks (like target nets) is important to the community, and understanding the types of representation which lead toward stable algorithms is useful scientifically.

A few weaknesses which currently prevent me from increasing my scoring---but I believe could be addressed easily during the discussion period--- [W1] There are a few highly related works whose citations were missed. [W2] The theoretical and empirical results do not support some of the conclusions drawn in the paper, especially that functional regularization is the cause of the improved performance. [W3] A few assumptions / assertions made during the motivation of the work seem unsupported (both in this paper and in the literature more broadly). [W4] The experiments have a few odd choices.

---

[W1] Two very important papers which should be cited and discussed:

1. Representations for Stable Off-policy Reinforcement Learning [Ghosh and Bellemare, 2020]
2. Improving Performance in Reinforcement Learning by Breaking Generalization in Neural Networks [Ghiassian et al., 2020]

The first heavily discusses the role of (learned) representations on stability in deep RL. It shows empirically and theoretically several properties of representations that yield provably convergent (thus stable according to their def'n) deep RL algorithms.

The second proposes a highly similar approach as this submission. Instead of learned fourier features being used to separate the data on the frequency domain, Ghiassian et al. separate the features spatially using tile-coding fed into a neural network. Ghiassian et al. have similar results showing that this spatial separation significantly reduces the need for target networks (and thus improves stability according to this submission's def'n).

---

[W2] A central conclusion (and the title of the paper) indicate that the cause of the improved performance is _functional regularization_. However, I do not believe that any empirical or theoretical results actually show that this is the case. For the theory results, the only claims that can be supported are that the network converges to specific frequencies faster than others, which is a bit different than showing a form of regularization (note that faster convergence does not imply a different final solution, nor a constraint on solution space). Because the proposed algorithm additionally concatenates the original features, the function class is actually strictly _larger_ for the proposed transformation and so the difference in performance _can't_ be functional  regularization.

I highly recommend reconsidering the title of the paper and the claims made throughout to more closely match what the paper can actually provide evidence for. Noise filtration, easier tuning of hyperparams, etc. all seem like more plausible and directly supportable claims.

---

[W3] This is a slight continuation of W2, but there are a few other instances where claims are not directly supported. During the motivation (roughly lines 31-39) the paper discusses weaknesses in prior approaches: dropout, weight-space regularization, early stopping. It isn't clear, however, that any of these approaches are actually harmful. Early stopping is perhaps intuitively harmful, but why is l1 or l2 regularization more harmful for bootstrapping than for the standard supervised prediction problem? This is less clearly true to me. The paper would benefit greatly by either (a) supporting these claims with citations or (b) supporting these claims empirically by comparing to a standard MLP + l2 regularization (for instance).

**edit:** After reading other reviews and checking the paper again, I notice that I had missed that Figure 5 does include a baseline with l2 regularization. The core of my point still stands, however, that there are several instances of not quite supported claims throughout the paper.

---

[W4] Some of the empirical decisions are not well motivated and may suggest some overclaiming and unfairness.

1. Why use 5 seeds and 68% confidence intervals? This implies that most of the results are 32% likely to be incorrect (and for 8 domains this means the first experiment has approx a 93% chance of at least one claim being wrong). And with 5 seeds, the assumption that variance is appropriately estimated (and so the confidence intervals are accurate) is a pretty strong assumption.
2. The statement on lines 247-248 appears totally incorrect. I count 4 wins (Cheetah, Quad Run, Quad Walk, Hopper), 3 ties (Acrobot, Finger, Walker) and 1 loss (Humanoid). The proposed still looks good considering these results, but the paper should be much more careful about avoiding overclaims.
3. The environment subselection is a little concerning. How did you pick these 8 domains from the approx 28 domains in DMControl? Similarly, how did you pick the 4 image based domains? Of the 4 chosen image domains, 3 were where the proposed algorithm performed best for state-based experiments. This appears to be biasing results in favor of the proposed algorithm via domain cherry-picking.

Finally, along the same lines of W2 and W3, section 6.2 is not directly measuring the conclusions it is trying to draw. By designing an experiment comparing two algorithms without target networks, you can make claims about the impact of target networks on performance. You cannot make claims about "stability" of the algorithm, nor can you support claims about the noise in the targets when not using target networks. If you want to make claims about stability and noise, you need to measure stability and noise.

That said, I actually believe section 6.2 has really interesting results about the interaction between your representation and the need for target networks. Keeping the same empirical results and changing the wording to be about target networks (instead of noise and stability) would make this section considerably stronger.

**Time Spent Reviewing:**

6

---

> ### Author Response · Authors · 2021-08-11
> **Response to Reviewer # N6ix**
>
> Dear Reviewer,
>
> Thank you for the helpful and insightful feedback! We respond to your comments below.
>
> > *[W1] Two very important papers which should be cited and discussed: [Ghosh and Bellemare, 2020] and [Ghiassian et al., 2020]*
> - Thank you! We will cite these and add the following discussion.
> - [Ghiassian et al., 2020] use tile coding to reduce catastrophic interference on small control problems. However, it’s unclear how this method will scale up to high-dimensional control problems or is able to generalize to in-distribution states that are nonetheless not seen during training. In contrast, LF2 works on environments as big as 78 state dimensions and 12 action dimensions (Quadruped), and is an easy plug-and-play replacement for MLPs, since the same hyperparameters do well across a wide range of environments. More importantly, both these methods are complementary and can be combined -- tile coding can be applied to the LF2 network input.
> - [Ghosh and Bellemare, 2020] examine when TD learning with fixed representations is stable. Although they provide interesting conditions for convergence, their assumptions are restrictive and are difficult to draw conclusions from for continuous MDPs with stochastic updates. Analyzing our proposed LF2 architecture under their mathematical framework is an interesting line for future work.
>
> > *[W2] For the theory results, the only claims that can be supported are that the network converges to specific frequencies faster than others, which is a bit different than showing a form of regularization (note that faster convergence does not imply a different final solution, nor a constraint on solution space).*
> - That’s a good point! However, in practice, it is very common to train with a fixed (reasonable) number of gradient steps. When using such a finite number of gradient updates, changing the per-frequency convergence rate *does* drastically change the final solution. We show this effect in Fig. 1 with 1D targets: LF2 networks with different sigma values (which controls the convergence rate) trained for a small, fixed number of epochs results in very different learned functions.
>
> > *[W2] Because the proposed algorithm additionally concatenates the original features, the function class is actually strictly larger for the proposed transformation and so the difference in performance can't be functional regularization.*
> - The original features have dimensionality between 7 and 90, which is much smaller than the Fourier feature dimension of 1024 that we use. The NTK kernel function with concatenation is a weighted combination of the Fourier feature kernel function and the 1-layer-shallower MLP kernel function. Due to the mismatch in dimensionality, the Fourier feature kernel function is much more heavily weighted. This means that the network’s convergence rate behavior is strongly driven by the Fourier features. We make a note of this on Line 175-177: “Note that concatenating x is omitted for this two-layer 176 model. This is because any contribution from concatenation goes to zero as we increase the layer width.” We will clarify this further.
>
> > *[W2] I highly recommend reconsidering the title of the paper and the claims made throughout to more closely match what the paper can actually provide evidence for. Noise filtration, easier tuning of hyperparams, etc. all seem like more plausible and directly supportable claims.*
> - Thank you for the suggestion -- we will be open to these changes. However, we believe that our use of the term “functional regularization” is consistent with other usage in the literature [1,2,3]. For example, Piché et.al. [1] use the term to refer to penalizing how much the Q-function deviates from the target network in function space, even though their method still has the same fixed point. Similarly, our method constrains how quickly our learned function changes in frequency space, even though the eventual fixed point should be the same (assuming a very large number of gradient steps).
>
> > *[W3] Why is l1 or l2 regularization more harmful for bootstrapping than for the standard supervised prediction problem? This is less clearly true to me.*
> - In the original submission, we actually compared MLP + L2 regularization in Figure 5, where it’s denoted as “MLP + weight decay.” (Note that weight decay in PyTorch’s Adam optimizer is exactly the same as L2 regularization). L2 generally helps a little, but not as much as our proposed method.
> - We believe that L1 and L2 regularization, which act in weight space, are still helpful but less effective than directly controlling the learned function. This is because it’s unclear how exactly a L1/L2 penalty changes the learning dynamics or controls underfitting/overfitting in different parts of the state-action space.
> - In the case of supervised learning, the learning dynamics are less important (as long as there are no exploding/vanishing gradients) because the intermediate network outputs aren’t used. In contrast, bootstrapping relies on intermediate network iterates to produce target values, so controlling the intermediate learned function, which LF2 does, is much more important.
>
> > *[W4] Why use 5 seeds and 68% confidence intervals?*
> - *Confidence*: Sorry for the terminology confusion here. The shaded confidence interval in the plots of our paper is *exactly* the “standard error of mean” which is a common way to show the error bars in machine learning  (e.g. in [4]). We will reword the phrase in the paper to avoid this confusion for readers.
> - *Seeds*: We run 5 seeds due to academic lab compute constraints. We ran a total of 170 final experiments with 5 seeds each, for a total of 850 runs. Each run took about 12hrs. This is a total of 10K+ GPU hours just for final experiments without including the experiments run during the development of the approach. We will try adding more seeds in the final version.
>
> > *[W4] The statement on lines 247-248 appears totally incorrect. I count 4 wins (Cheetah, Quad Run, Quad Walk, Hopper), 3 ties (Acrobot, Finger, Walker) and 1 loss (Humanoid).*
> - Thanks for pointing it out -- we will fix this. We had been looking only at LF2 vs vanilla MLPs while making the claim, but will change it to reflect LF2 performance vs the best baseline in each separate environment. LF2 is still around the top for every environment, and is compatible with weight decay and the other regularization methods we added at the beginning of the discussion phase, so performance may further increase when LF2 is combined with these methods.
>
> > *[W4] How did you pick these 8 domains from the approx 28 domains in DMControl? Similarly, how did you pick the 4 image based domains?*
> - State-based: Due to our limited compute budget, we decided to not use DMControl environments that are too easy and are basically fully solved by any method. Instead, we picked the environments which are commonly evaluated in RL papers and are present in both DMControl as well as OpenAI Gym -- i.e. Cheetah, Walker, Hopper, Acrobot, Quadruped, and Humanoid.
> - Image-based: SAC is much slower to train for image-based methods. Hence, out of the above 8, we picked the four *moderately difficult* environments, discarding the ones which were too easy from state-space. We also left out Humanoid because no image-based method we tried could make progress on it (note: most DMControl image-based RL papers don’t show Humanoid). We are currently evaluating more image-based environments and will post results once the experiments finish.
>
> > *[W4] By designing an experiment comparing two algorithms without target networks, you can make claims about the impact of target networks on performance. You cannot make claims about "stability" of the algorithm, nor can you support claims about the noise in the targets when not using target networks. If you want to make claims about stability and noise, you need to measure stability and noise.*
> - Thanks for the suggestion! However, it is difficult to measure the noise in the target values, as (a) it is unclear what the ground truth target values should be in the middle of training, and (b) the Q-functions in deep RL do not tend to converge to the same values by the end of training. In the paper, we attempt to simulate increased noise by removing the target networks.
> - We do agree with you that this doesn’t exactly prove that LF2 is more stable -- removing the target network changes more than just faster propagation of noisy target values. To specifically test the hypothesis that LF2 does better because of improved stability properties, we ran a new experiment where we add zero-mean Gaussian noise to the target values. The result is here: [https://drive.google.com/file/d/1rwLTr-AyY10yo4D3mBULQuey5JLqHLnm/view?usp=sharing](https://drive.google.com/file/d/1rwLTr-AyY10yo4D3mBULQuey5JLqHLnm/view?usp=sharing) .
> - For this experiment, we keep target networks for both LF2 and MLPs. We see that LF2 generally does much better than MLPs, and can maintain decent performance even under significant amounts of added noise (e.g. standard deviation of 30). This experiment precisely shows that LF2 tends to be more stable than MLPs, and we can extrapolate from this to infer that stability is at least partially why LF2 does better than MLPs when no noise is artificially added.
>
> ### References
> [1] Beyond Target Networks: Improving Deep Q-learning with Functional Regularization, Alexandre Piché, Joseph Marino, Gian Maria Marconi, Christopher Pal, Mohammad Emtiyaz Khan, 2021
>
> [2] Pan, P., Swaroop, S., Immer, A., Eschenhagen, R., Turner, R. E., & Khan, M. E. (2020). Continual deep learning by functional regularisation of memorable past. arXiv preprint arXiv:2004.14070.
>
> [3] Titsias, M. K., Schwarz, J., Matthews, A. G. D. G., Pascanu, R., & Teh, Y. W. (2019). Functional regularisation for continual learning with gaussian processes. arXiv preprint arXiv:1901.11356.
>
> [4] Deep Reinforcement Learning that Matters, Henderson et al. 2017

---

### Author Response · Authors · 2021-08-11
**[To All] Brief summary of rebuttal**


We thank the reviewers for their valuable feedback. To summarize, we propose a simple architecture for off-policy deep RL based on learned Fourier features. Tuning the initialization variance of the Fourier features controls the rate at which the network fits signals of different frequencies, which alleviates the effect of noise in the bootstrap process.

We’re pleased to report that we’ve *finished* the additional baselines or experiments suggested by the reviewers. We made detailed responses directly to each review. Here, we recap the main points from reviewers and the resulting experimental findings:
- **More baselines [R3 # aHKm, R4 # Susc]**: We additionally run dropout, spectral normalization, and functional regularization on the state-space environments. We find that dropout and functional regularization help with sample efficiency, though LF2 consistently does well compared to these new baselines. We also run L2 regularization on the image-based environments and find that it only hurts performance. State-based results are in [this link](https://drive.google.com/file/d/1iaIiTHX6T93ZV17SMYRrYaL6VgkIaBX5/view?usp=sharing), and image-based results are in [this link](https://drive.google.com/file/d/1coSuLkYR9dr1JbLYrwNCcz3IeWddYrZS/view?usp=sharing).
- **Stability of LF2 under noise [R1 # N6ix, R3 # aHKm, R4 # Susc]**: Reviewers were unsure whether we proved that LF2 does better because it fits noise more slowly. We run an experiment where we add Gaussian noise to the bootstrap targets and compare LF2 and MLP in this setting. We find that LF2 is far more robust than MLP and maintains surprising levels of performance even when significant amounts of noise is artificially added. Results are in [this link](https://drive.google.com/file/d/1rwLTr-AyY10yo4D3mBULQuey5JLqHLnm/view?usp=sharing).
- **Is input concatenation responsible for the LF2 performance increase? [R3 # aHKm]**: We concatenate the input to the first layer output of a vanilla MLP and evaluate it for state-space SAC. We find that this does not improve performance. Results are in [this link](https://drive.google.com/file/d/1j98y78nP5U6tKJt1TtQ2R4vgzI5cOuEr/view?usp=sharing).
- **New theoretical analysis of LF2 convergence [R2 # jV7a]**: As suggested by reviewer jV7a, we use the framework from [1] to examine the stability of LF2 under Bellman updates. The analysis shows that Bellman updates are always contractions once we use large enough sigma. However, this result mainly applies to discrete MDPs, not the continuous ones we mainly explore in our paper. The analysis is provided in [this link](https://drive.google.com/file/d/1EXYky8xoOtbdISiJ5_mLrGXg4IReih04/view?usp=sharing).

For full details, please see the responses to individual reviews.

---

### Decision · Program_Chairs · 2021-09-27

**Decision:**

Accept (Poster)

**Comment:**

This paper considered the learned Fourier features in RL as an alternative parametrization to the vanilla MLP for Q network. The author argued that the benefits of the Fourier features can **i)** reduces high-frequency noise; **ii)** the noise is the major performance killer in RL. Then, the authors showed the better performances of the proposed parametrized Q in off-policy RL tasks on DeepMind Control Suite from both state and image.

Besides the novelty of using Fourier features raised by several reviewers (jV7a, aHKm,  Susc),  the major issue of this paper is that the claimed benefits of Fourier feature layer are indeed the main reason leading to better performances (N6iX and Susc). Specifically, the empirical study does not provide convincing support of claim, while the theoretical justification does not offer strong insight (Reviewer N6ix) and actually may not able to explain the empirical results.

 - As reviewer Susc mentioned, the motivation is clearly justified: **i)** reduces noise; **ii)** the noise is the major performance killer in RL.


 - The difference between vanilla MLP and Fourier feature is in fact different parametrization families. With square loss in Eq. 4, the optimal solution actually is $E[y|x]$ if the parametrization is realizable. All parametrization is actually able to reduce the noise. How can the author guarantee the superior comes from uncertainty canceling, not **approximation error** reduction is never discussed.


 - In the theoretical analysis part, the author calculate NTK, which relies on the assumption in the **limit case** and the neural network weights are not far away from initialization. In the experiment, only around 1000 random feature is used. As many recent papers [1, 2 and to name a few] suggest the situation is different in finite case. With this gap in mind, it is not clear to me the phenomena from kernel can be straightforwardly extend to random feature. To support the claim, the author may first compare the NTK vs. random feature.


- The authors also provide the contraction proof for kernel feature. **i)** this proof is not clear to me whether the relu layer is used; **ii)** in fact, how to characterize the benefits of the contractition vs. noise cancellation should be discussed.

Let me quote the suggestion from Reviewer Susc "validating all hypotheses should be a main feature of papers like this one, not just showing improvements." As a scientific paper, we should make careful claim with supportive evidence. The paper indeed reveals some interesting phenomena, but with some overclaims without convincing support.

Minor:

The term "functional regularization" might not be appropriate and not aligned with its vanilla meaning. In this paper, it is in fact parametrization family selection.

Since most of the reviewers agreed to accept this paper, I will still recommend acceptance for this submission. However, I would suggest the authors can revise the paper,  taking the comments in reviews and meta-review into account. Specifically, reduce the over-simplified justification and the misleading claim, just make the claims which the empirical evidence can support to avoid the potential misleading of the whole community.

[1] Fort, Stanislav, Gintare Karolina Dziugaite, Mansheej Paul, Sepideh Kharaghani, Daniel M. Roy, and Surya Ganguli. "Deep learning versus kernel learning: an empirical study of loss landscape geometry and the time evolution of the neural tangent kernel." arXiv preprint arXiv:2010.15110 (2020).

[2] Allen-Zhu, Zeyuan, and Yuanzhi Li. "What can ResNet learn efficiently, going beyond kernels?." arXiv preprint arXiv:1905.10337 (2019).